# Global Convergence of Adaptive Sensing for Principal Eigenvector Estimation

**Alex Saad-Falcon** [1]   **Brighton Ancelin** [1]   **Justin Romberg** [1]

## Abstract

Principal component analysis classically requires full $d$-dimensional samples, yet in various applications hardware limits acquisition to a few scalar measurements per sample. We analyze a compressed variant of Oja's algorithm for estimating the principal eigenvector of the data covariance matrix using only two adaptive measurements per sample. At each iteration, we observe one measurement along the current estimate and one in a random orthogonal direction. We prove that after $t$ iterations, the expected sine-squared error to the true eigenvector is $\mathcal{O}(\lambda_1\lambda_2 d^2/(\Delta^2 t))$, where $d$ is the ambient dimension, $\lambda_1, \lambda_2$ are the leading eigenvalues, and $\Delta = \lambda_1 - \lambda_2$ is the eigengap. We complement this with a matching information-theoretic lower bound of $\Omega(\lambda_1\lambda_2 d^2/(\Delta^2 t))$ — the first for compressed eigenvector estimation — proving that the $d^2$ factor, an additional factor of $d$ compared to the fully-observed minimax rate $\Theta(\lambda_1\lambda_2 d/(\Delta^2 t))$, is the fundamental cost of compression and cannot be improved. In contrast, any non-adaptive scheme with two measurements per iteration suffers $\Omega(\lambda_2^2 d^3/(\Delta^2 t))$, an additional power of $d$. This separates fully-observed, adaptive-compressed, and non-adaptive-compressed PCA across three powers of $d$. Our analysis handles the noisy setting where the covariance has nonzero trailing eigenvalues, providing the first convergence guarantee for adaptive compressed subspace tracking beyond the noiseless case.

## 1. Introduction

Principal component analysis (PCA) is a fundamental technique for identifying low-dimensional structure in high-dimensional data. While batch methods like randomized

SVD (Halko et al., 2011) compute principal components efficiently, they require storing or repeatedly accessing the full dataset. Streaming algorithms like Oja's method (Oja, 1982) overcome this limitation by updating the subspace estimate one sample at a time, requiring only $O(d)$ memory.

Both approaches, however, assume access to full $d$-dimensional samples. This presents a significant limitation in settings where acquiring complete samples is physically infeasible — e.g., radar, wireless communications, genomics, MRI, and recommender systems. In this *compressed sensing regime*, we observe only a few scalar measurements per sample rather than the full vector.

The two-measurement streaming setting arises from either physical or practical constraints in several domains where full-dimensional samples are intractable to attain and/or store. In phased-array radar and mmWave communications, modern hybrid-beamforming architectures pair large antenna arrays with far fewer RF chains: typical 5G mmWave designs use only a handful of RF chains per array, with each chain producing a single scalar measurement per time slot (Heath et al., 2016). In neural signal processing, implantable recording arrays must compress extracellular signals on-chip before wireless transmission, motivating a body of work on sensor-side compressed sensing for high-channel-count neural interfaces (Sun & Zhao, 2021). In wideband cognitive-radio spectrum sensing, periodic sub-Nyquist sampling estimates the power spectrum of a wide-sense stationary signal directly from compressed samples, bypassing signal reconstruction at the Nyquist rate (Ariananda & Leus, 2012). In all three settings, the data is observed once through a physical measurement process and cannot be revisited, so streaming algorithms with provable convergence guarantees are essential.

Existing analyses of compressed subspace tracking assume noiseless data where observations lie exactly in the signal subspace (i.e., $\lambda_2 = \cdots = \lambda_d = 0$). In our "noisy" setting, observations contain variance from trailing eigenvalues ($\lambda_2 > 0$), which acts as noise when estimating the signal direction $\bar{u}$. No prior work has established information-theoretic lower bounds for adaptive compressed sensing with a fixed number of measurements per iteration.

We develop convergence guarantees for a compressed variant of Oja's algorithm using only two adaptive measure-

---

[1]School of Electrical and Computer Engineering, Georgia Institute of Technology, Atlanta, GA, USA. Correspondence to: Alex Saad-Falcon <alexsaadfalcon@gatech.edu>.

*Proceedings of the 43$^{rd}$ International Conference on Machine Learning*, Seoul, South Korea. PMLR 306, 2026. Copyright 2026 by the author(s).

ments per iteration. At each step, we take one measurement along the current estimate and another in a random orthogonal direction, balancing exploitation with exploration.

Our main result establishes matching upper and lower bounds:

**Theorem 1.1. (Informal)** *Let* $\Sigma$ *have leading eigenvalues* $\lambda_1 > \lambda_2$ *with eigengap* $\Delta = \lambda_1 - \lambda_2$. *For Gaussian data* $\boldsymbol{v}_t \sim \mathcal{N}(\boldsymbol{0}, \Sigma)$ *observed through two adaptive linear measurements per iteration:*

**Upper bound:** *Algorithm 1 achieves:*

$$\mathbb{E}[1 - (\bar{\boldsymbol{u}}^T \boldsymbol{u}_t)^2] = \mathcal{O}(\lambda_1 \lambda_2 d^2 / (\Delta^2 t)) \qquad (1)$$

**Lower bound:** *Any estimator satisfies:*

$$\mathbb{E}[1 - (\bar{\boldsymbol{u}}^T \hat{\boldsymbol{u}}_t)^2] = \Omega(\lambda_1 \lambda_2 d^2 / (\Delta^2 t)) \qquad (2)$$

This establishes the minimax rate $\Theta(\lambda_1 \lambda_2 d^2 / (\Delta^2 t))$ for adaptive compressed PCA with two measurements per iteration. Compared to fully-observed PCA with rate $\Theta(\lambda_1 \lambda_2 d / (\Delta^2 t))$, our rate includes an additional factor of $d$ — the fundamental cost of observing only a 2-dimensional projection at each iteration. The key contributions of this work include:

- The first convergence analysis for adaptive compressed PCA with noisy observations, achieving the optimal $\Theta(\lambda_1 \lambda_2 d^2 / (\Delta^2 t))$ rate.

- A matching information-theoretic lower bound via Assouad's lemma, proving the $d^2$ factor is unavoidable.

- A novel Assouad-based derivation of the classical $\Omega(\lambda_1 \lambda_2 d / (\Delta^2 t))$ lower bound for fully-observed PCA.

- Fixed-point analysis for tracking time-varying subspaces.

The lower bound proof (Appendix B) exploits a *measurement budget* argument: two unit-norm measurements per iteration provide total "energy" $2t$ that must cover $d-1$ coordinates where the eigenvector could vary, giving average energy $O(t/d)$ per coordinate and limiting per-coordinate estimation precision. The averaged form of Assouad's lemma aggregates this difficulty across coordinates, yielding the $\Omega(\lambda_1 \lambda_2 d^2 / (\Delta^2 t))$ bound.

The paper is organized as follows. Section 2 reviews related work. Section 3 formalizes the problem and presents Algorithm 1. Section 4 presents the upper bound, lower bound, and tracking analysis. Section 5 sketches the proof techniques. Section 6 provides experimental validation, and Section 7 discusses limitations and extensions to rank-$k$ subspaces.

## 2. Related Work

### 2.1. Streaming PCA and Subspace Tracking

Subspace tracking algorithms iteratively update the subspace estimate as new data arrives, requiring only $O(dk)$ memory for rank-$k$ estimation. Oja's algorithm (Oja, 1982) performs a rank-one update followed by normalization at each iteration, with convergence guarantees developed in (Balsubramani et al., 2013; Allen-Zhu & Li, 2017; Huang et al., 2021; Zhang & Balzano, 2016b). The algorithm is equivalent to projected gradient descent on the Grassmann manifold (Balzano, 2022).

### 2.2. Randomized and Batch PCA

When data can be accessed in multiple passes, randomized methods achieve near-optimal complexity for approximating the *sample* covariance or its singular value decomposition. Randomized SVD (Halko et al., 2011) computes a rank-$k$ approximation using $O(k)$ passes over the data, with extensions to single-pass streaming via sketching (Woodruff, 2014). Liberty's Frequent Directions algorithm (Liberty, 2013) provides deterministic streaming guarantees with optimal $O(dk/\epsilon)$ space for $(1 + \epsilon)$-approximate sample covariance. Block Krylov methods (Musco & Musco, 2015) achieve optimal $\tilde{O}(k/\sqrt{\epsilon})$ matrix-vector product complexity. These methods treat the observed matrix as the object of interest, providing worst-case guarantees without distributional assumptions. In contrast, Oja's algorithm and its variants operate under a statistical model where columns are drawn i.i.d. from a distribution with covariance $\Sigma$, and the goal is to estimate the *population* eigenvectors of $\Sigma$—a fundamentally different objective that enables finite-sample statistical rates but requires stochastic assumptions.

These methods share a key limitation: they require access to full $d$-dimensional samples. In settings where acquiring complete samples is infeasible — e.g., large sensor arrays, distributed systems, or bandwidth-limited channels — compressed measurement schemes become necessary. Our work addresses precisely this gap.

### 2.3. Convergence Bounds for Streaming PCA

A detailed survey of bounds for Oja's algorithm can be found in (Allen-Zhu & Li, 2017). Prior work on fully-observed streaming PCA develops upper bounds of the form

$$\mathbb{E}[1 - (\bar{\boldsymbol{u}}^T \boldsymbol{u}_t)^2] \leq \tilde{\mathcal{O}}\left(\frac{\lambda_1 \lambda_2}{\Delta^p} \cdot \frac{d^q}{t}\right),$$

where $\tilde{\mathcal{O}}$ suppresses logarithmic factors. Under sub-Gaussian assumptions, (Li et al., 2018) obtains $p = 2$ and $q = 1$.

The minimax lower bound from (Vu & Lei, 2013) states that

for any estimator $\boldsymbol{u}_t$ based on $t$ full-dimensional samples,

$$\inf_{\boldsymbol{u}_t} \sup_{P \in \mathcal{M}(\lambda_1, \lambda_2, d)} \mathbb{E}[1 - (\bar{\boldsymbol{u}}^T \boldsymbol{u}_t)^2] = \Omega \left( \frac{\lambda_1 \lambda_2}{\Delta^2} \cdot \frac{d}{t} \right).$$

Our work extends this to the compressed setting, establishing that the minimax rate with two measurements per iteration is $\Theta(\lambda_1 \lambda_2 d^2 / (\Delta^2 t))$.

### 2.4. Compressive Measurements and Adaptive Sensing

Several works extend subspace tracking to settings with missing or compressively sampled data (Balzano et al., 2018; Zhang & Balzano, 2016a; Gonen et al., 2016). The adaptive setting — where measurements depend on the current estimate — has been studied in (Ongie et al., 2017; 2018), but these analyses assume noiseless data ($\lambda_2 = 0$). Interestingly, (Gonen et al., 2016) proves that adaptive sensing achieves strictly better sample complexity than non-adaptive random measurements.

There are few upper bound results applicable without full observations (Huang et al., 2021; De Sa et al., 2015), and none that simultaneously handle adaptive measurements and noise. Our work fills this gap, providing the first convergence analysis for adaptive compressed PCA with noisy observations.

Compressed sensing arises naturally in many domains: radar and communications systems where hardware constraints limit simultaneous channel acquisition (Costanzo et al., 2016), genomics where measuring all gene expressions is costly (Xavier et al., 2017), MRI where scan time must be minimized (Jaspan et al., 2015), and recommender systems where users rate only a subset of items (Gogna & Majumdar, 2015). In array processing applications such as radar, sonar, and wireless communications, adaptive sensing enables deployment of large sensor arrays with fewer active channels, improving angular resolution without proportional increases in system complexity (Van Trees, 2002). Our formulation in terms of eigenvalues and the eigengap applies directly to these settings.

## 3. Problem Setup

We study principal eigenvector estimation in a streaming setting where samples $\boldsymbol{v}_t \sim \mathcal{N}(\boldsymbol{0}, \boldsymbol{\Sigma})$ arrive sequentially and cannot be stored. Instead of observing the full vector $\boldsymbol{v}_t \in \mathbb{R}^d$, we obtain only two linear measurements $\boldsymbol{x}_t = \boldsymbol{A}_t \boldsymbol{v}_t$ where $\boldsymbol{A}_t \in \mathbb{R}^{2 \times d}$. The challenge is to recover the principal eigenvector $\bar{\boldsymbol{u}}$ from these compressed measurements using only the current sample and estimate $\boldsymbol{u}_t$ at each iteration.

When the measurement matrices $\boldsymbol{A}_t$ are chosen randomly and independently of the current estimate, they are generally misaligned with both the true principal eigenvector and the

---

**Algorithm 1** Compressive Oja's, Rank One, Adaptive Sensing

1: **Initialize**: $\boldsymbol{u}_0 \in \mathbb{R}^d, \|\boldsymbol{u}_0\|_2 = 1$
2: **Step Size**: $\eta_t > 0$
3: **for** $t \in 1 \dots T$ **do**
4:     **Draw sample:** $\boldsymbol{v}_t \in \mathbb{R}^d \sim \mathcal{N}(\boldsymbol{0}, \boldsymbol{\Sigma})$
5:     **Draw random unit vector:**
    $\boldsymbol{b}_t \in \mathbb{R}^d, \|\boldsymbol{b}_t\|_2 = 1, \boldsymbol{b}_t \perp \boldsymbol{u}_t$
6:     **Compressive measurement:**
    $\boldsymbol{x}_t = \boldsymbol{A}_t \boldsymbol{v}_t = [\boldsymbol{u}_t^T \boldsymbol{v}_t; \boldsymbol{b}_t^T \boldsymbol{v}_t]$
7:     **Update eigenvector:**
    $\hat{\boldsymbol{u}}_{t+1} = \boldsymbol{u}_t + \eta_t (\boldsymbol{u}_t \boldsymbol{u}_t^T + \boldsymbol{b}_t \boldsymbol{b}_t^T) \boldsymbol{v}_t \boldsymbol{v}_t^T \boldsymbol{u}_t$
8:     **Normalize:** $\boldsymbol{u}_{t+1} = \hat{\boldsymbol{u}}_{t+1} / \|\hat{\boldsymbol{u}}_{t+1}\|_2$
9: **end for**
10: **Return** $\boldsymbol{u}_T$

---

current estimate $\boldsymbol{u}_t$. This misalignment introduces a critical challenge: randomly oriented measurements provide limited information about the signal direction. When data is drawn from a covariance matrix $\boldsymbol{\Sigma}$ with leading eigenvalue $\lambda_1$ much larger than trailing eigenvalues $\lambda_2, \dots, \lambda_d$, most of the variance lies along the principal eigenvector $\bar{\boldsymbol{u}}$. Random measurements, however, capture a mixture of signal (the principal component) and noise (the trailing eigenspace), with no guarantee that the signal component is well-represented. This becomes particularly severe in high dimensions when random measurement vectors $\boldsymbol{A}_t$ are nearly orthogonal to the principal direction $\bar{\boldsymbol{u}}$.

Adaptive sensing addresses this limitation by designing the measurement matrix $\boldsymbol{A}_t$ based on the current estimate $\boldsymbol{u}_t$. A natural approach would measure only along the current estimate, i.e., setting $\boldsymbol{A}_t = \boldsymbol{u}_t^T$. However, this would only reinforce our current estimate without capturing information from orthogonal directions needed to correct errors or track changes. The key insight is to balance exploitation (measuring along $\boldsymbol{u}_t$) with exploration (measuring in a direction orthogonal to $\boldsymbol{u}_t$), ensuring both signal amplification and sufficient exploration to enable convergence.

### 3.1. Algorithm

We focus on estimating the leading eigenvector $\bar{\boldsymbol{u}}$ of the covariance matrix using only two measurements per iteration, i.e., $\boldsymbol{A}_t \in \mathbb{R}^{2 \times d}$. Given a current eigenvector estimate $\boldsymbol{u}_t$, we use the adaptive sampling scheme proposed in (Ongie et al., 2017) which chooses:

$$\boldsymbol{A}_t = \begin{bmatrix} \boldsymbol{u}_t^T \\ \boldsymbol{b}_t^T \end{bmatrix} \qquad \boldsymbol{x}_t = \boldsymbol{A}_t \boldsymbol{v}_t$$

where $\boldsymbol{b}_t$ is a unit vector drawn randomly from the orthogonal complement of $\boldsymbol{u}_t$. $\boldsymbol{b}_t$ can be constructed by taking $\hat{\boldsymbol{b}}_t \sim \mathcal{N}(0, \boldsymbol{I} - \boldsymbol{u}_t \boldsymbol{u}_t^T)$ and normalizing $\boldsymbol{b}_t = \hat{\boldsymbol{b}}_t / \|\hat{\boldsymbol{b}}_t\|_2$.

We assume normally distributed data with arbitrary covariance $\boldsymbol{v}_t \sim \mathcal{N}(\mathbf{0}, \boldsymbol{\Sigma})$. We denote the principal eigenvector of $\boldsymbol{\Sigma}$ as $\bar{\boldsymbol{u}}$, its eigenvalues as $\lambda_1 > \lambda_2 \geq \lambda_3 \geq \cdots \geq \lambda_d$, and its eigengap as $\Delta = \lambda_1 - \lambda_2$. Importantly, our formulation treats "noise" as part of the data-generating distribution (the trailing eigenspace) rather than additive measurement corruption — the leading eigenvector $\bar{\boldsymbol{u}}$ represents the "signal" direction we aim to estimate.

*Remark* 3.1 (Connection to Spiked Covariance Model). Our eigenvalue formulation generalizes the classical *spiked covariance model* $\boldsymbol{v}_t = \bar{\boldsymbol{u}} s_t + \boldsymbol{e}_t$ with $s_t \sim \mathcal{N}(0, 1)$ and $\boldsymbol{e}_t \sim \mathcal{N}(\mathbf{0}, \sigma^2 \mathbf{I})$, which yields $\boldsymbol{\Sigma} = \bar{\boldsymbol{u}}\bar{\boldsymbol{u}}^T + \sigma^2 \mathbf{I}$ with $\lambda_1 = 1 + \sigma^2$, $\lambda_2 = \sigma^2$, and $\Delta = 1$. Our formulation also handles arbitrary covariance structures with multiple distinct trailing eigenvalues.

The algorithm update in line 7 of Algorithm 1 can be derived from the Adaptive GROUSE framework (Ongie et al., 2017) by imputing the high-dimensional observation from compressed measurements (see Appendix A.1 for details). The imputed data simplifies to the projection $\tilde{\boldsymbol{v}}_t = (\boldsymbol{u}_t \boldsymbol{u}_t^T + \boldsymbol{b}_t \boldsymbol{b}_t^T)\boldsymbol{v}_t$, which is then used in the standard Oja update. The key innovation of our work is *not* the algorithm design itself, but rather the *proof technique* used to establish convergence guarantees in the noisy setting. Beyond the upper bound, we contribute three additional novel results: (i) a new Assouad-based proof of the classical $\Omega(\lambda_1 \lambda_2 d/(\Delta^2 t))$ lower bound for fully-observed PCA, providing an alternative to existing information-theoretic arguments; (ii) the first $\Omega(\lambda_1 \lambda_2 d^2/(\Delta^2 t))$ lower bound for compressed PCA, proving that the $d^2$ factor in our upper bound is optimal; and (iii) fixed-point analysis characterizing the steady-state tracking error when the principal eigenvector drifts over time.

## 4. Main Results

We now state our main convergence guarantees. Our upper bound establishes that Algorithm 1 converges in two phases: a warmup phase from random initialization, followed by a local convergence phase where the error decays as $\mathcal{O}(1/t)$.

**Theorem 4.1.** (Upper Bound) *Let $\boldsymbol{u}_0 \in \mathbb{R}^d$ be a random unit vector drawn uniformly from the unit sphere $\mathcal{S}^{d-1}$. Consider the data generation model $\boldsymbol{v}_t \sim \mathcal{N}(\mathbf{0}, \boldsymbol{\Sigma})$ where $\boldsymbol{\Sigma}$ has top eigenvalue $\lambda_1$ in direction $\bar{\boldsymbol{u}}$, all remaining eigenvalues at most $\lambda_2$, and eigengap $\Delta = \lambda_1 - \lambda_2 > 0$. Define:*

$$S = \frac{3\lambda_1 \lambda_2 d^2}{\Delta^2} + \frac{15\lambda_1 d}{\Delta}$$

*The following convergence guarantees hold:*

*(i)* **Warmup Phase:** *When Algorithm 1 is applied with step size $\eta_0 = (d-1)/(2S\Delta)$, then after $t_0$ iterations the expected squared sine alignment satisfies*

$\mathbb{E}[1 - (\bar{\boldsymbol{u}}^T \boldsymbol{u}_{t_0})^2] \leq 0.5$, *where:*

$$t_0 = (4S + 1) \log\left(\frac{d}{2}\right)$$

*(ii)* **Local Convergence Phase:** *For all $t \geq t_0$ applying the step size:*

$$\eta_t = \frac{2(d-1)}{\Delta(4S + (t - t_0))}$$

*the expected squared sine alignment decays at the following rate:*

$$\mathbb{E}[1 - (\bar{\boldsymbol{u}}^T \boldsymbol{u}_t)^2] \leq \frac{C_1}{4S + (t - t_0)} + \frac{C_2}{(4S + (t - t_0))^2}$$

*with constants:*

$$C_1 = 4S + 2 \qquad C_2 = \frac{(4S + 1)^2}{2}$$

For $t - t_0 \gg 4S$ (sufficiently many iterations past warmup), the dominant term in the Theorem 4.1 bound simplifies as

$$\frac{C_1}{4S + (t - t_0)} \approx \frac{4S}{t} = \frac{12\,\lambda_1 \lambda_2 d^2}{\Delta^2 t} + \frac{60\,\lambda_1 d}{\Delta\,t}.$$

The leading term is $\mathcal{O}(\lambda_1 \lambda_2 d^2/(\Delta^2 t))$, and the second term $\mathcal{O}(\lambda_1 d/(\Delta\,t))$ is lower order whenever $\lambda_2 d/\Delta \gg 5$.[1] Compared to the fully-observed rank-1 PCA setting with minimax rate $\Theta(\lambda_1 \lambda_2 d/(\Delta^2 t))$, our rate carries an extra factor of $d$, arising because two scalar measurements only resolve a two-dimensional subspace per iteration.

The result extends to additive measurement noise: if the compressed observation is corrupted as $\boldsymbol{x}_t = \boldsymbol{A}_t \boldsymbol{v}_t + \vec{\varepsilon}_t$ with $\vec{\varepsilon}_t \sim \mathcal{N}(\mathbf{0}, \sigma_\varepsilon^2 \mathbf{I}_2)$, then since $\boldsymbol{A}_t \boldsymbol{A}_t^T = \mathbf{I}_2$, the effective covariance seen by the algorithm becomes $\boldsymbol{\Sigma} + \sigma_\varepsilon^2 \mathbf{I}$, which inflates every eigenvalue by $\sigma_\varepsilon^2$ while preserving the eigengap $\Delta$. All bounds therefore hold with $\lambda_2$ replaced by $\lambda_2 + \sigma_\varepsilon^2$, which increases $S$ and slows convergence.

A key contribution of this work is proving that the $d^2$ factor in our upper bound is *optimal*:

**Theorem 4.2.** (Lower Bound) *Let $\mathcal{G}(\lambda_1, \lambda_2, d)$ denote the family of Gaussian distributions $\mathcal{N}(\mathbf{0}, \boldsymbol{\Sigma})$ in $\mathbb{R}^d$ where $\boldsymbol{\Sigma}$ has top eigenvalue $\lambda_1$ in some direction $\bar{\boldsymbol{u}}$, all remaining eigenvalues at most $\lambda_2$, and eigengap $\Delta = \lambda_1 - \lambda_2 > 0$. For any estimator $\hat{\boldsymbol{u}}_t$ based on $t$ i.i.d. samples from $P$, each observed through any two linear measurements, the minimax risk satisfies:*

$$\inf_{\hat{\boldsymbol{u}}_t} \sup_{P \in \mathcal{G}(\lambda_1, \lambda_2, d)} \mathbb{E}\left[1 - (\bar{\boldsymbol{u}}^T \hat{\boldsymbol{u}}_t)^2\right] = \Omega\left(\frac{\lambda_1 \lambda_2 d^2}{\Delta^2 t}\right)$$

---

[1]The constants in $S$ reflect a Cauchy–Schwarz bound on the general-$\Sigma$ self-term and an Isserlis correction to its variance; the rate order $\Theta(\lambda_1 \lambda_2 d^2/(\Delta^2 t))$ is unaffected.

*More precisely, for $t \geq \frac{(d-1)^2 \lambda_1 \lambda_2}{36 \Delta^2}$:*

$$\inf_{\hat{\boldsymbol{u}}_t} \sup_{P \in \mathcal{G}(\lambda_1, \lambda_2, d)} \mathbb{E}\left[1 - (\bar{\boldsymbol{u}}^T \hat{\boldsymbol{u}}_t)^2\right] \geq \frac{(d-1)^2 \lambda_1 \lambda_2}{864 \Delta^2 t}$$

*Proof.* See Appendix B for the complete proof using Assouad's lemma. □

**Proof Sketch.** The proof uses Assouad's lemma, which reduces minimax estimation to distinguishing $2^{d-1}$ hypotheses indexed by hypercube vertices $v \in \{-1, +1\}^{d-1}$. Each hypothesis corresponds to a covariance $\boldsymbol{\Sigma}_v = \lambda_2 \mathbf{I} + \Delta \bar{\boldsymbol{u}}_v \bar{\boldsymbol{u}}_v^T$ (a rank-one perturbation of $\lambda_2 \mathbf{I}$, which lies inside $\mathcal{G}(\lambda_1, \lambda_2, d)$) with principal eigenvector:

$$\bar{\boldsymbol{u}}_v = \boldsymbol{e}_1 \sqrt{1 - \frac{(d-1)\beta^2}{4}} + \frac{\beta}{2} \sum_{j=1}^{d-1} v_j \boldsymbol{e}_{j+1}$$

where $\beta$ controls the perturbation magnitude. Adjacent hypotheses (differing in one coordinate) have eigenvectors with sine-squared distance $\alpha^2 = \beta^2 - \beta^4/4$.

The key insight is a *measurement budget* argument. Two unit-norm measurement vectors per iteration provide total "energy" 2, so over $t$ iterations the cumulative energy across all $d-1$ perturbation coordinates is at most $2t$ — giving average energy $O(t/d)$ per coordinate. Since the KL divergence for distinguishing hypotheses in coordinate $j$ scales with the energy allocated to that coordinate, the averaged total variation across coordinates is correspondingly limited via Pinsker and Jensen. The averaged form of Assouad's lemma aggregates these per-coordinate contributions across all $d-1$ coordinates, yielding the $\Omega(\lambda_1 \lambda_2 d^2/(\Delta^2 t))$ lower bound. See Appendix B for the complete proof.

Together, Theorems 4.1 and 4.2 establish that the minimax rate for adaptive compressed sensing with two measurements per iteration is $\Theta(\lambda_1 \lambda_2 d^2/(\Delta^2 t))$ on $\mathcal{G}(\lambda_1, \lambda_2, d)$. The matching is order-optimal: the upper bound's dominant term $12\lambda_1 \lambda_2 d^2/(\Delta^2 t)$ exceeds the lower bound's $(d-1)^2 \lambda_1 \lambda_2/(864\Delta^2 t)$ by a factor of $12 \cdot 864 = 10{,}368$ as $d \to \infty$. To our knowledge, this is the first information-theoretic lower bound for compressed sensing estimation of the principal eigenvector. As a byproduct, our Assouad-based proof technique also yields a novel derivation of the classical $\Omega(\lambda_1 \lambda_2 d/(\Delta^2 t))$ lower bound for fully-observed PCA (Appendix B, Equation (17)), providing an alternative to existing Fano-based proofs (Vu & Lei, 2013).

**Non-Adaptive Compression Penalty.** When the measurement matrices $\boldsymbol{A}_1, \ldots, \boldsymbol{A}_t$ are fixed *before* observing any data, the minimax rate becomes strictly harder. Appendix B establishes a non-adaptive lower bound of $\Omega(\lambda_2^2 d^3/(\Delta^2 t))$

(Equation (19)), an additional power of $d$ beyond the adaptive bound (18). The three regimes scale as $d^1$, $d^2$, $d^3$: fully-observed Oja, adaptive compressed Oja (one $d$ from the $m = 2$ measurement bottleneck), and non-adaptive compressed Oja (a second $d$ from worst-case orientation). Quantitatively, the projected eigengap shrinks from $\Delta$ to $\Delta/d$, giving a per-sample Cramér–Rao penalty of $d^2$; the energy budget further restricts informative measurements to a fraction $\sim 1/d$, yielding the net factor of $d$. In the hard regime $\Delta \leq \lambda_2$ this matches the adaptive eigenvalue scaling $\lambda_1 \lambda_2$ up to a factor of 2 (Corollary B.4), giving $\Omega(\lambda_1 \lambda_2 d^3/(\Delta^2 t))$.

**Comparison to Prior Work.** The GROUSE analysis of (Zhang & Balzano, 2016b) provides "with high probability" bounds in the exactly low-rank case ($\lambda_2 = 0$) and "in expectation" monotonic-improvement bounds in the noisy case, leaving global convergence under noise to future work; the Adaptive GROUSE analysis of (Ongie et al., 2017) provides "with high probability" bounds (built on per-step monotonic-improvement-in-expectation lemmas), again assuming $\lambda_2 = 0$. This noiseless assumption greatly simplifies the analysis by eliminating cross-terms between signal and noise components that would otherwise proliferate in the stochastic recurrence relations.

In contrast, our setting with noisy observations drawn from $\boldsymbol{v}_t \sim \mathcal{N}(\mathbf{0}, \boldsymbol{\Sigma})$ introduces fundamental challenges. The adaptive coupling between algorithm iterates (where the current estimate $\boldsymbol{u}_t$ determines the measurement matrix $\boldsymbol{A}_t$) combined with signal-noise interactions makes concentration-based analysis difficult. Our proof instead uses stochastic recurrence relations to track the expected error at each step, carefully accounting for both the adaptive dependencies and the signal-noise interactions. This approach yields direct "in expectation" bounds of the form $\mathbb{E}[\text{error}] \leq \epsilon$, avoiding the nested probabilistic structure of "with high probability in expectation" statements while handling the technical challenges introduced by measurement noise.

Three specific components in the upper-bound analysis (Appendix A.2.5) enable the noisy guarantee, each absent from or trivialized by the noiseless ($\lambda_2 = 0$) GROUSE analyses:

1. *Signal/noise decomposition for general covariance*: The conditional covariance $\mathbb{E}[gh \mid c, z] = \boldsymbol{u}^T \boldsymbol{\Sigma} \boldsymbol{b}$ is bounded by Cauchy–Schwarz, $(\boldsymbol{u}^T \boldsymbol{\Sigma} \boldsymbol{b})^2 \leq (\boldsymbol{u}^T \boldsymbol{\Sigma} \boldsymbol{u})(\boldsymbol{b}^T \boldsymbol{\Sigma} \boldsymbol{b}) \leq a^2 b^2$ with $a^2 = \Delta c^2 + \lambda_2$ and $b^2 = \Delta z^2 + \lambda_2$; via Isserlis's theorem this contributes a $2a^2 b^2$ term to the self-term variance that must be tracked through the recurrence (Appendix A.2.3).

2. *Self-term bound via monotonicity + Jensen*: Bounding $\mathbb{E}[c^2 X^2/(X^2 + Y^2) \mid c, z]$ where $X = 1 + \eta g^2$ and

$Y = \eta gh$ are correlated combines monotonicity in $X$ (using $X \geq 1$) with Jensen's inequality applied to the convex map $1/(1 + y)$, sidestepping the matrix-concentration arguments that the adaptive coupling between $\boldsymbol{A}_t$ and $\boldsymbol{u}_t$ would complicate.

3. *Integration over the exploration direction*: The random orthogonal direction $\boldsymbol{b}_t$ on the unit sphere has known law $\mathbb{E}[z^2 \mid c] = (1 - c^2)/(d - 1)$, and the conditional variance $b^2 = \Delta z^2 + \lambda_2$ depends on $z$. The explicit $\mathbb{E}_z[\cdot]$ integration is the source of the $1/(d - 1)$ factor in the recurrence.

No component is individually novel; the combination is what extends the matched $\Theta(\lambda_1 \lambda_2 d^2/(\Delta^2 t))$ rate to noisy data. The corresponding lower-bound novelty is the measurement-energy-budget argument behind the $d^2$ (adaptive) and $d^3$ (non-adaptive) scalings.

### 4.1. Extension: Tracking a Moving Eigenvector

Our stochastic recurrence framework extends naturally to the non-stationary setting where the underlying eigenvector $\bar{\boldsymbol{u}}_t$ changes over time. While this extension is not our main contribution, it demonstrates the flexibility of our analysis approach and addresses an important application scenario — e.g., adaptive beamforming and direction of arrival estimation ((Van Trees, 2002), Chapters 7, 9).

Assuming the eigenvector drift satisfies $(\bar{\boldsymbol{u}}_t^T \bar{\boldsymbol{u}}_{t+1})^2 \geq 1 - V$ for some maximum "velocity" $V$, we show in Appendix A.2.11 that with a constant learning rate, the algorithm achieves a steady-state error of:

$$x^* = V + \sqrt{VS} \qquad (3)$$

at the optimal step size:

$$\hat{\eta} = \sqrt{\frac{V}{S}} \qquad (4)$$

Section 6 validates this result empirically (Figure 3).

In the next section, we sketch the key ideas behind these results.

## 5. Proof Techniques

This section sketches the key ideas behind Theorem 4.1. Our analysis proceeds in three stages: (1) we decompose the per-iteration update into "self-term" and "cross-term" contributions, (2) we derive bounds on each term and combine them into a stochastic recurrence on the expected cosine alignment, and (3) we analyze this recurrence separately in the warmup and local convergence phases. Full details appear in Appendix A.2.5.

With two unit-norm measurements per iteration, the total "measurement energy" available after $t$ iterations is $2t$, spread by pigeonhole across the $d - 1$ orthogonal coordinates of the residual. Average energy per coordinate is therefore $O(t/d)$, so the per-coordinate squared error decays as $O(d/t)$, and summing over $d - 1$ coordinates yields $\Theta(d^2/t)$. Theorem 4.2 formalizes this argument via Assouad's lemma in Appendix B.

To examine the convergence of Algorithm 1, we begin with the algorithm update rule:

$$\hat{\boldsymbol{u}}_{t+1} = \boldsymbol{u}_t + \eta(\boldsymbol{u}_t \boldsymbol{u}_t^T + \boldsymbol{b}_t \boldsymbol{b}_t^T)\boldsymbol{v}_t \boldsymbol{v}_t^T \boldsymbol{u}_t$$

$$\boldsymbol{u}_{t+1} = \frac{\hat{\boldsymbol{u}}_{t+1}}{\|\hat{\boldsymbol{u}}_{t+1}\|_2}$$

We define a few auxiliary variables, dropping subscripts $t$ for brevity:

$$c = \bar{\boldsymbol{u}}^T \boldsymbol{u} \qquad z = \bar{\boldsymbol{u}}^T \boldsymbol{b}$$

$$g = \boldsymbol{v}^T \boldsymbol{u} \qquad h = \boldsymbol{v}^T \boldsymbol{b}$$

Geometrically, $c$ is the cosine alignment between the current estimate $\boldsymbol{u}$ and the true eigenvector $\bar{\boldsymbol{u}}$, while $z$ measures how much the random exploration direction $\boldsymbol{b}$ happens to overlap with $\bar{\boldsymbol{u}}$. The scalars $g$ and $h$ are the two compressed measurements — projections of the data sample onto $\boldsymbol{u}$ and $\boldsymbol{b}$ respectively. We will implicitly be conditioning on the current iterate $\boldsymbol{u}$ every iteration, which makes $c$ a constant. Note that:

$$g \sim \mathcal{N}(0, \boldsymbol{u}^T \boldsymbol{\Sigma} \boldsymbol{u}) \qquad h|\boldsymbol{b} \sim \mathcal{N}(0, \boldsymbol{b}^T \boldsymbol{\Sigma} \boldsymbol{b})$$

$$\mathbb{E}[gh|\boldsymbol{b}] = \boldsymbol{u}^T \boldsymbol{\Sigma} \boldsymbol{b}$$

which are derived in Appendix A.2.1 and Appendix A.2.2. Notably, the (squared) cosine alignment $c^2$ will be the core quantity we aim to bound over the course of iterations $t$. We can compute the pieces of the update as:

$$\hat{\boldsymbol{u}}_{t+1} = \boldsymbol{u} + \eta(\boldsymbol{u}\boldsymbol{u}^T + \boldsymbol{b}\boldsymbol{b}^T)\boldsymbol{v}\boldsymbol{v}^T \boldsymbol{u}$$
$$= \boldsymbol{u} + \eta\boldsymbol{u}\boldsymbol{u}^T \boldsymbol{v}\boldsymbol{v}^T \boldsymbol{u} + \eta\boldsymbol{b}\boldsymbol{b}^T \boldsymbol{v}\boldsymbol{v}^T \boldsymbol{u}$$
$$= \boldsymbol{u}(1 + \eta g^2) + \boldsymbol{b}(\eta gh)$$

$$\bar{\boldsymbol{u}}^T \hat{\boldsymbol{u}}_{t+1} = c(1 + \eta g^2) + z(\eta gh)$$

$$\|\hat{\boldsymbol{u}}_{t+1}\|_2^2 = (1 + \eta g^2)^2 + (\eta gh)^2$$

We then combine these into a recurrence on the cosine alignment:

$$c_{t+1}^2 = \frac{(\bar{\boldsymbol{u}}^T \hat{\boldsymbol{u}}_{t+1})^2}{\|\hat{\boldsymbol{u}}_{t+1}\|_2^2} = \frac{(c(1 + \eta g^2) + z(\eta gh))^2}{(1 + \eta g^2)^2 + (\eta gh)^2} \qquad (5)$$

To simplify, we define two scalar random variables and take expectations conditioned on the current iterate $c$:

$$X = 1 + \eta g^2 \qquad Y = \eta g h$$

$$\mathbb{E}[c_{t+1}^2 \mid c] = \mathbb{E}_{z,X,Y}\left[\frac{(cX + zY)^2}{X^2 + Y^2}\bigg| c\right]$$

Dropping the nonnegative $z^2$ term leaves us with:

$$\mathbb{E}[c_{t+1}^2 \mid c] \geq \mathbb{E}\left[c^2\frac{X^2}{X^2 + Y^2} + 2cz\frac{XY}{X^2 + Y^2}\bigg| c\right]$$

We refer to the first term as the "self-term" and the second as the "cross-term."

### 5.1. Self-term Bound

All of the expectations are first over $g$ and $h$ (or $X$ and $Y$) conditioned on $c$ and $z$, leaving the expectation over $z$ for last. This is helpful, since this conditioning results in $g$ and $h$ being jointly normal. The conditional covariance $\mathbb{E}[gh \mid c, z] = \boldsymbol{u}^T\boldsymbol{\Sigma}\boldsymbol{b}$ satisfies $(\boldsymbol{u}^T\boldsymbol{\Sigma}\boldsymbol{b})^2 \leq (\boldsymbol{u}^T\boldsymbol{\Sigma}\boldsymbol{u})(\boldsymbol{b}^T\boldsymbol{\Sigma}\boldsymbol{b}) \leq a^2b^2$ by Cauchy–Schwarz, so via Isserlis's theorem it contributes a $2(\boldsymbol{u}^T\boldsymbol{\Sigma}\boldsymbol{b})^2 \leq 2a^2b^2$ term beyond the product $\mathbb{E}[g^2]\mathbb{E}[h^2]$. We derive the following self-term bound in Appendix A.2.3:

$$\mathbb{E}\left[c^2\frac{X^2}{X^2 + Y^2}\right] \geq \mathbb{E}_z\left[\frac{c^2}{1 + 3\eta^2a^2b^2}\right]$$

where:

$$\mathbb{E}[g^2] \leq a^2 := \Delta c^2 + \lambda_2 \qquad \mathbb{E}[h^2] \leq b^2 := \Delta z^2 + \lambda_2$$

### 5.2. Cross-term Bound

The derivation of this bound is in Appendix A.2.4.

$$\mathbb{E}\left[2cz\frac{XY}{X^2 + Y^2}\right] \geq \mathbb{E}_z\left[\frac{2\eta\Delta c^2 z^2}{1 + 3\eta(a^2 + b^2)}\right]$$

### 5.3. Combining Bounds and Constructing Recurrence

We put together these two bounds to get a lower bound on the expected cosine similarity of the next iterate given the current iterate:

$$\mathbb{E}[c_{t+1}^2|c] \geq \mathbb{E}\left[c^2\frac{X^2}{X^2 + Y^2} + 2cz\frac{XY}{X^2 + Y^2}\bigg| c\right]$$
$$\geq \mathbb{E}_z\left[\frac{c^2}{1 + 3\eta^2a^2b^2}\bigg| c\right]$$
$$+ \mathbb{E}_z\left[\frac{2\eta\Delta c^2 z^2}{1 + 3\eta(a^2 + b^2)}\bigg| c\right]$$

$$a^2 = \Delta c^2 + \lambda_2 \qquad b^2 = \Delta z^2 + \lambda_2$$

We define an auxiliary step size that absorbs problem-dependent constants, simplifying the recurrence analysis:

$$\hat{\eta} = \frac{\Delta\eta}{d - 1}$$

The step sizes in Theorem 4.1 are stated in terms of $\eta$; converting via $\eta = (d - 1)\hat{\eta}/\Delta$ recovers the theorem statement. And simplify the recurrence in Appendix A.2.5 to get:

$$\mathbb{E}[c_{t+1}^2|c] \geq c^2 + 2\hat{\eta}c^2(1 - c^2) - Sc^2\hat{\eta}^2 \qquad (6)$$

$$S = \frac{3\lambda_1\lambda_2 d^2}{\Delta^2} + \frac{15d\lambda_1}{\Delta}$$

The factor 3 in the leading term of $S$ comes from the Cauchy–Schwarz bound on the Isserlis self-term. The constant 15 in the second term collapses $z^2$-dependent contributions from both the self- and cross-terms via $\Delta + \lambda_2 = \lambda_1$ (see Appendix A.2.5 for details). For the convergence analysis, it is more convenient to work with the *error* rather than the alignment. We define $x_t = \mathbb{E}[1 - c_t^2]$, the expected sine-squared distance to the true eigenvector. Since $c_t^2 \to 1$ corresponds to convergence, we equivalently want $x_t \to 0$. Taking complements in Equation 6 yields the recurrence:

$$x_{t+1} \leq x_t(1 - 2\hat{\eta}(1 - x_t)) + S\hat{\eta}^2 \qquad (7)$$

We analyze this recurrence in two phases.

### 5.4. Warmup Phase

The warmup phase covers $c_t^2 \in [\epsilon, 0.5]$, where $\epsilon = 1/d$ for random initialization. We pick a constant step size $\hat{\eta}_0 = 1/2S$ that ensures monotonic improvement while not regressing once $c_t^2$ reaches 0.5. With this step size, we show in Appendix A.2.6 that:

$$\mathbb{E}[c_{t+1}^2|c] \geq c^2(1 + \frac{1}{4S})$$

Telescoping gives:

$$\mathbb{E}\left[c_{t_0}^2\bigg| c_0\right] \geq \epsilon(1 + \frac{1}{4S})^{t_0}$$

Setting this $\geq 0.5$ and solving for $t_0$ yields:

$$t_0 \geq (4S + 1)\log(\frac{d}{2}) \qquad (8)$$

### 5.5. Local Convergence Phase

For the local convergence phase with $x_t \leq 0.5$, Equation 7 simplifies to:

$$x_{t+1} \leq x_t(1 - \hat{\eta}) + S\hat{\eta}^2 \qquad (9)$$

*Table 1.* Iterations to reach target error $10^{-2}$ for adaptive vs. non-adaptive measurements ($\lambda_1 = 2$, $\lambda_2 = 1$, learning rate $\eta = 0.01/d$, median of 20 trials). Both methods use exactly two scalar measurements per iteration. The Slowdown column is computed from the unrounded medians; dividing the displayed (rounded) values differs slightly in the last digit.

| Dimension $d$ | Adaptive | Non-Adaptive | Slowdown |
|---|---|---|---|
| 16 | $3.8 \times 10^4$ | $1.6 \times 10^5$ | $4.2\times$ |
| 32 | $1.9 \times 10^5$ | $1.3 \times 10^6$ | $7.1\times$ |
| 64 | $8.4 \times 10^5$ | $1.2 \times 10^7$ | $14\times$ |

We choose a decaying step size $\hat{\eta}_t = K/(T+t)$ with $K = 2$ and $T = 4S$. Unrolling this recurrence (details in Appendix A.2.8), we obtain:

$$x_t \leq \frac{C_1}{T+t} + \frac{x_0 C_2}{(T+t)^K} \qquad (10)$$

with constants $C_1 = 4S + 2$ and $C_2 = (4S+1)^2$. (The form stated in Theorem 4.1 absorbs the post-warmup initial condition $x_{t_0} = 1/2$ into the second coefficient, giving the equivalent $C_2 = (4S+1)^2/2$ used there.) Combining Equations 8 and 10 completes the proof of Theorem 4.1.

## 6. Experiments

We validate Theorems 4.1 and 4.2 empirically by simulating Algorithm 1. Unless otherwise stated, experiments use $\lambda_1 = 2$, $\lambda_2 = 1$ (eigengap $\Delta = 1$) and the decaying step-size schedule of Theorem 4.1.

### 6.1. Adaptive vs. Non-Adaptive Measurements

A natural question is whether the adaptive measurement strategy — measuring along the current estimate $\boldsymbol{u}_t$ and its orthogonal complement — is necessary, or whether non-adaptive random measurements suffice. Table 1 shows that adaptivity provides a substantial, dimension-dependent advantage.

The slowdown grows with dimension: from $4.2\times$ at $d = 16$ to $14\times$ at $d = 64$. This has an intuitive explanation: a random 2-dimensional subspace has expected squared overlap of only $O(1/d)$ with any fixed direction, so non-adaptive sensing extracts little useful gradient information per iteration. In contrast, our adaptive strategy measures along the current estimate $\boldsymbol{u}_t$, creating a positive feedback loop where any existing alignment with the signal is amplified.

### 6.2. Upper Bound, Lower Bound, and Empirical Error

Figure 1 jointly shows our upper and lower bounds together with averaged empirical error trajectories. The empirical median tracks the upper and lower bounds to within constant factors across the local-convergence phase. The upper

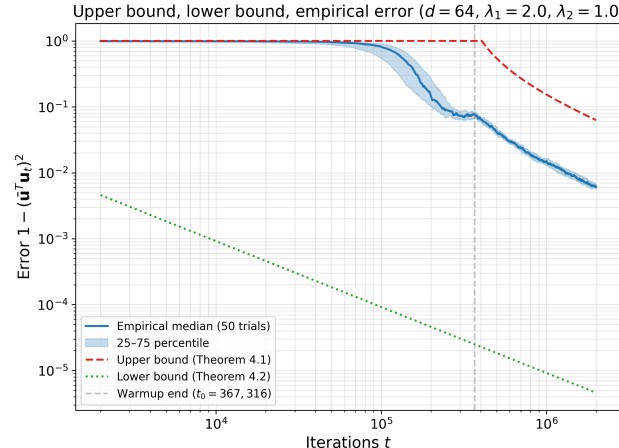

*Figure 1.* Upper bound (Theorem 4.1), minimax lower bound (Theorem 4.2), and empirical error of Algorithm 1 on shared axes. Empirical curve shows median (solid) and 25–75 percentile band (shaded) over 50 trials. The constant-factor gap between the two bounds is $12 \cdot 864 = 10{,}368$ asymptotically (Section 4); the empirical curve sits neatly between the two bounds. Parameters: $d = 64$, $\lambda_1 = 2$, $\lambda_2 = 1$, step sizes from Theorem 4.1.

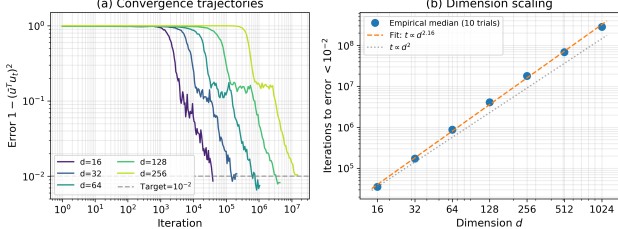

*Figure 2.* Empirical validation of the $d^2$ dimension scaling. (a) Convergence trajectories at $d \in \{16, 32, 64, 128, 256\}$, each median crossing the target error $10^{-2}$ (dashed gray) at progressively later iterations. (b) Log-log iterations to reach target error vs. dimension on the extended grid $d \in \{16, 32, 64, 128, 256, 512, 1024\}$ with 10 trials per dimension; the empirical least-squares fit $t \propto d^{2.16}$ (orange dashed) closely tracks the theoretical leading-order $t \propto d^2$ (gray dotted). Simulations use $\lambda_1 = 2$, $\lambda_2 = 1$, learning rate $\eta = 0.01/d$.

bound is tighter than the lower bound, which is typical of Assouad-based proofs, where the lower-bound construction is intrinsically conservative.

### 6.3. Dimension Scaling

To validate our theoretical prediction that convergence time scales as $\Theta(d^2)$, we run Algorithm 1 across dimensions $d \in \{16, 32, 64, 128, 256, 512, 1024\}$ and measure iterations to reach target error $10^{-2}$. Figure 2(a) shows convergence trajectories at five dimensions reaching the $10^{-2}$ target; Figure 2(b) shows the log-log scaling analysis on the extended seven-dimension grid, and Table 2 reports the corresponding numerical values.

The empirical scaling exponent is 2.16 on $d \in$

*Table 2.* Numerical results for the dimension scaling experiment. Iterations to reach target error $10^{-2}$ (median of 10 trials per dimension) with $\lambda_1 = 2$, $\lambda_2 = 1$, learning rate $\eta = 0.01/d$. The ratio column shows $t_d/t_{d/2}$, which equals 4 for exact $d^2$ scaling.

| Dimension $d$ | Iterations | Ratio |
|---|---|---|
| 16 | 35,500 | — |
| 32 | 172,750 | 4.87 |
| 64 | 879,190 | 5.09 |
| 128 | 4,091,830 | 4.65 |
| 256 | 17,950,000 | 4.39 |
| 512 | 68,500,000 | 3.82 |
| 1024 | 284,650,000 | 4.15 |

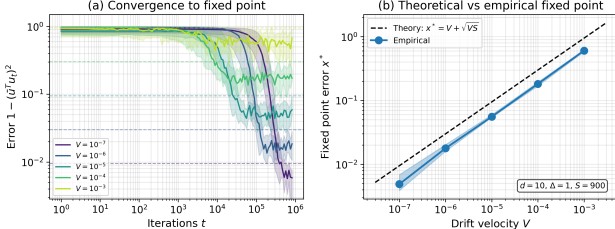

*Figure 3.* Validation of tracking analysis. (a) Convergence trajectories for drift velocities $V \in \{10^{-7}, 10^{-6}, 10^{-5}, 10^{-4}, 10^{-3}\}$, each converging to a different steady-state error. Solid lines show medians over 20 trials with 20–80 percentile shading; dashed lines show theoretical fixed points from Equation 3. (b) Theoretical fixed point $x^* = V + \sqrt{VS}$ (dashed) versus empirical steady-state error (solid with 20–80 percentile shading). Parameters: $d = 10$, $\lambda_1 = 2$, $\lambda_2 = 1$, step size from Equation 4, initial alignment $(\bar{\boldsymbol{u}}^T \boldsymbol{u}_0)^2 = 0.1$.

$\{16, 32, 64, 128, 256, 512, 1024\}$ with 10 trials per dimension, closely matching the theoretical $d^2$ prediction. Two finite-sample effects account for the small excess over 2: (i) the second term $15\lambda_1 d/\Delta$ in $S = 3\lambda_1\lambda_2 d^2/\Delta^2 + 15\lambda_1 d/\Delta$ is non-negligible at moderate $d$, contributing an $O(d)$ component to the leading-order rate; and (ii) the warmup phase $t_0 = (4S + 1)\log(d/2)$ adds a $\log d$ correction. The leading $d^2$ term dominates as $d \to \infty$, so the fitted exponent approaches 2 asymptotically.

The ratio between successive dimensions in Table 2 hovers around 4–5, consistent with $d^2$ scaling; Appendix C provides further validation, including eigengap dependence.

### 6.4. Tracking Validation

Figure 3 validates the tracking analysis from Section 4.1, simulating Algorithm 1 with a time-varying principal eigenvector that rotates by a random angle $\theta$ per iteration with $\sin^2(\theta) = V$ at the constant step size of Equation 4. Panel (a) shows the error converging to a steady-state value that increases with drift velocity, and Panel (b) confirms that the theoretical fixed point $x^* = V + \sqrt{VS}$ provides a valid

upper bound across four orders of magnitude in $V$.

## 7. Limitations and Future Work

Our convergence bound's quadratic dependence on dimension $d$ is optimal for algorithms using two measurements per iteration (Appendix B): with only two scalar observations per sample, $\Omega(d)$ iterations are required to distinguish hypotheses across all coordinates.

Extending our results to rank-$k$ PCA is a natural direction. The fully-observed streaming rate for rank-$k$ PCA matches the Vu–Lei minimax lower bound (Vu & Lei, 2013) up to a logarithmic factor, $\widetilde{\mathcal{O}}(\lambda_k\lambda_{k+1} k(d-k)/(\Delta_k^2 t))$ with $\Delta_k = \lambda_k - \lambda_{k+1}$, established for Oja's subspace iteration by (Liang et al., 2023). By analogy with our rank-1 results, we conjecture that taking $m$ measurements per sample (natural choice $m = 2k$) incurs a multiplicative *dimension penalty* of $d/m$ in the adaptive case and $(d/m)^2$ in the non-adaptive case. The main obstacle to formalizing this is the orthonormalization step needed to maintain a valid $k$-dimensional subspace estimate, which couples the $k$ stochastic recurrences and blocks direct application of the rank-1 analysis (Balsubramani et al., 2013).

Our analysis also extends to sub-Gaussian data. Recent work on fully-observed Oja's algorithm (Li et al., 2018; Liang, 2023) extends convergence guarantees to sub-Gaussian distributions with the same $\lambda_1\lambda_2/\Delta^2$ scaling. Our proof uses Gaussianity only via Isserlis's theorem for the upper-bound fourth moments (replaceable by sub-Gaussian moment bounds) and the closed-form Gaussian KL divergence in the lower-bound construction (replaceable by Le Cam with $\chi^2$-divergence); the energy-budget argument behind the $d^2$ scaling is itself distribution-free, so the result extends to sub-Gaussian data via these substitutions.

Extending the (non-compressive) sparse PCA literature (Vu & Lei, 2013; Cai et al., 2013) to compressive measurements is also open.

## Acknowledgements

This work was supported in part by CogniSense, one of the seven centers sponsored by the Semiconductor Research Corporation (SRC) and DARPA under the Joint University Microelectronics Program 2.0 (JUMP 2.0).

## Impact Statement

This paper presents work whose goal is to advance the field of Machine Learning. There are many potential societal consequences of our work, none which we feel must be specifically highlighted here.

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

# A. Technical Appendices and Supplementary Material

## A.1. Imputation Derivation

To handle compressive measurements, we use the modification from (Balzano, 2022): impute the high-dimensional observation $\boldsymbol{v}_t$ to compute how much lies within our current estimate $\boldsymbol{u}_t$. We compute the projection weight $w_t$, the projected data $\boldsymbol{p}_t$, and the residual $\boldsymbol{r}_t$. This gives us:

$$w_t = (\boldsymbol{A}_t \boldsymbol{u}_t)^{\dagger} \boldsymbol{x}_t \qquad \boldsymbol{p}_t = \boldsymbol{u}_t w_t$$

$$\boldsymbol{r}_t = \boldsymbol{A}_t^T (\boldsymbol{x}_t - \boldsymbol{A}_t \boldsymbol{u}_t w_t) \qquad \tilde{\boldsymbol{v}}_t = \boldsymbol{u}_t w_t + \boldsymbol{r}_t$$

If we substitute values for the adaptive sensing setting, we get:

$$w_t = (\boldsymbol{A}_t \boldsymbol{u}_t)^{\dagger} \boldsymbol{x}_t = \begin{bmatrix} \boldsymbol{u}_t^T \boldsymbol{u}_t \\ \boldsymbol{b}_t^T \boldsymbol{u}_t \end{bmatrix}^{\dagger} \begin{bmatrix} \boldsymbol{u}_t^T \boldsymbol{v}_t \\ \boldsymbol{b}_t^T \boldsymbol{v}_t \end{bmatrix}$$

$$= [1, 0] \begin{bmatrix} \boldsymbol{u}_t^T \boldsymbol{v}_t \\ \boldsymbol{b}_t^T \boldsymbol{v}_t \end{bmatrix} = \boldsymbol{u}_t^T \boldsymbol{v}_t$$

$$\begin{aligned} \tilde{\boldsymbol{v}}_t &= \boldsymbol{u}_t w_t + \boldsymbol{r}_t \\ &= \boldsymbol{u}_t (\boldsymbol{A}_t \boldsymbol{u}_t)^{\dagger} \boldsymbol{x}_t + \boldsymbol{A}_t^T (\boldsymbol{x}_t - \boldsymbol{A}_t \boldsymbol{u}_t w_t) \\ &= \boldsymbol{u}_t \begin{bmatrix} 1, 0 \end{bmatrix} \begin{bmatrix} \boldsymbol{u}_t^T \boldsymbol{v}_t \\ \boldsymbol{b}_t^T \boldsymbol{v}_t \end{bmatrix} + [\boldsymbol{u}_t, \boldsymbol{b}_t] \left( \begin{bmatrix} \boldsymbol{u}_t^T \boldsymbol{v}_t \\ \boldsymbol{b}_t^T \boldsymbol{v}_t \end{bmatrix} - \begin{bmatrix} 1 \\ 0 \end{bmatrix} \boldsymbol{u}_t^T \boldsymbol{v}_t \right) \\ &= \boldsymbol{u}_t \boldsymbol{u}_t^T \boldsymbol{v}_t + (\boldsymbol{u}_t \boldsymbol{u}_t^T + \boldsymbol{b}_t \boldsymbol{b}_t^T) \boldsymbol{v}_t - \boldsymbol{u}_t \boldsymbol{u}_t^T \boldsymbol{v}_t \\ &= (\boldsymbol{u}_t \boldsymbol{u}_t^T + \boldsymbol{b}_t \boldsymbol{b}_t^T) \boldsymbol{v}_t \end{aligned}$$

This expression simplifies the imputed data $\tilde{\boldsymbol{v}}_t$ into the projection of the $d$-dimensional sample $\boldsymbol{v}_t$ onto the 2-dimensional subspace spanned by $\boldsymbol{u}_t$ and $\boldsymbol{b}_t$. The lift from $\mathbb{R}^2$ back to $\mathbb{R}^d$ uses $\boldsymbol{A}_t^T$ (the transpose, not $\boldsymbol{A}_t$ itself); since $\boldsymbol{A}_t$ has orthonormal rows, $\boldsymbol{A}_t^T \boldsymbol{A}_t = \boldsymbol{u}_t \boldsymbol{u}_t^T + \boldsymbol{b}_t \boldsymbol{b}_t^T$ is the orthogonal projector onto $\mathrm{span}(\boldsymbol{u}_t, \boldsymbol{b}_t)$. Thus the lift does *not* recover the full ambient-space residual — it recovers the residual's projection onto the measurement subspace, imputing zero outside it (the standard imputation construction of (Balzano, 2022)). The subsequent analysis works directly with this projected update and does not assume recovery of the full ambient residual. With non-adaptive measurements, the projection does not have such a simple form, primarily due to the structure of $(\boldsymbol{A}_t \boldsymbol{u}_t)^{\dagger}$ for random $\boldsymbol{A}_t$.

We can substitute this into Oja's full-dimensional update equation to get compressive Oja's with adaptive sensing:

$$\hat{\boldsymbol{u}}_{t+1} = \boldsymbol{u}_t + \eta_t (\boldsymbol{u}_t \boldsymbol{u}_t^T + \boldsymbol{b}_t \boldsymbol{b}_t^T) \boldsymbol{v}_t (\boldsymbol{v}_t^T \boldsymbol{u}_t)$$

where $w_t = \boldsymbol{v}_t^T \boldsymbol{u}_t$ is the projection weight observed in $\boldsymbol{x}_t$. We then normalize $\boldsymbol{u}_{t+1} = \hat{\boldsymbol{u}}_{t+1} / \|\hat{\boldsymbol{u}}_{t+1}\|_2$ to keep the estimate as a unit vector. The full algorithm is stated in Algorithm 1.

## A.2. Supporting Lemmas

### A.2.1. VARIANCE OF $g$ AND $h$

All expectations are taken conditioned on the current iterate $\boldsymbol{u}$, which makes $c = \bar{\boldsymbol{u}}^T \boldsymbol{u}$ a constant. We also just need upper bounds on each.

$$\mathbb{E}[g^2] = \mathbb{E}[(\boldsymbol{v}^T \boldsymbol{u})^2] = \mathbb{E}[\boldsymbol{u}^T \boldsymbol{v} \boldsymbol{v}^T \boldsymbol{u}] = \boldsymbol{u}^T \boldsymbol{\Sigma} \boldsymbol{u}$$

we can express the quadratic form using the eigendecomposition of $\boldsymbol{\Sigma}$ (which is positive semidefinite):

$$
\boldsymbol{u}^T \boldsymbol{\Sigma} \boldsymbol{u} = \sum_{i=1}^{d} \lambda_i (\boldsymbol{u}^T \boldsymbol{w}_i)^2
$$

$$
= \lambda_1 (\boldsymbol{u}^T \bar{\boldsymbol{u}})^2 + \sum_{i=2}^{d} \lambda_i (\boldsymbol{u}^T \boldsymbol{w}_i)^2
$$

$$
\leq \lambda_1 c^2 + \lambda_2 \sum_{i=2}^{d} (\boldsymbol{u}^T \boldsymbol{w}_i)^2
$$

$$
= \lambda_1 c^2 + \lambda_2 (1 - c^2)
$$

$$
= \Delta c^2 + \lambda_2 := a^2
$$

Where $\boldsymbol{w}_i$ is the $i$-th eigenvector of $\boldsymbol{\Sigma}$. Similarly for $h$ conditioning on $\boldsymbol{b}$, which makes $z = \bar{\boldsymbol{u}}^T \boldsymbol{b}$ a constant:

$$
\mathbb{E}[h^2 | \boldsymbol{b}] = \mathbb{E}[(\boldsymbol{v}^T \boldsymbol{b})^2 | \boldsymbol{b}] = \boldsymbol{b}^T \boldsymbol{\Sigma} \boldsymbol{b}
$$

$$
= \lambda_1 (\boldsymbol{b}^T \bar{\boldsymbol{u}})^2 + \sum_{i=2}^{d} \lambda_i (\boldsymbol{b}^T \boldsymbol{w}_i)^2
$$

$$
\leq \lambda_1 z^2 + \lambda_2 \sum_{i=2}^{d} (\boldsymbol{b}^T \boldsymbol{w}_i)^2
$$

$$
\leq \lambda_1 z^2 + \lambda_2 (1 - z^2)
$$

$$
= \Delta z^2 + \lambda_2 := b^2
$$

### A.2.2. COVARIANCE OF $g$ AND $h$

The covariance of $g$ and $h$ can easily be computed by first conditioning on $\boldsymbol{b}$:

$$
\mathbb{E}[gh] = \mathbb{E}_{\boldsymbol{b}}[\mathbb{E}[(\boldsymbol{u}^T \boldsymbol{v} \boldsymbol{v}^T \boldsymbol{b}) | \boldsymbol{b}]] = \mathbb{E}_{\boldsymbol{b}}[\boldsymbol{u}^T \boldsymbol{\Sigma} \boldsymbol{b}]
$$

We can notice that for the distribution of $\boldsymbol{b}$, the densities at $\boldsymbol{b}$ and $-\boldsymbol{b}$ are equal due to symmetry, so this gives us:

$$
\mathbb{E}[gh] = \mathbb{E}_{\boldsymbol{b}}[\boldsymbol{u}^T \boldsymbol{\Sigma} \boldsymbol{b}] = 0
$$

For the cross-term, we actually need:

$$
\mathbb{E}[cz * gh] = c\mathbb{E}_{\boldsymbol{b}}[\mathbb{E}[(\bar{\boldsymbol{u}}^T \boldsymbol{b})(\boldsymbol{b}^T \boldsymbol{v} \boldsymbol{v}^T \boldsymbol{u}) | \boldsymbol{b}]]
$$

$$
= c\mathbb{E}_{\boldsymbol{b}}[\bar{\boldsymbol{u}}^T \boldsymbol{b} \boldsymbol{b}^T \boldsymbol{\Sigma} \boldsymbol{u}]
$$

We know since $\boldsymbol{b}$ is a unit vector drawn from the orthogonal complement of $\boldsymbol{u}$, we have:

$$
\mathbb{E}[\boldsymbol{b} \boldsymbol{b}^T] = \frac{1}{d-1}(\mathbf{I} - \boldsymbol{u} \boldsymbol{u}^T)
$$

Plugging this in, we get:

$$
\mathbb{E}[cz * gh] = \frac{c}{d-1}(\bar{\boldsymbol{u}}^T (\mathbf{I} - \boldsymbol{u} \boldsymbol{u}^T) \boldsymbol{\Sigma} \boldsymbol{u})
$$

$$
= \frac{c}{d-1}((\bar{\boldsymbol{u}}^T \boldsymbol{\Sigma} \boldsymbol{u}) - (\bar{\boldsymbol{u}}^T \boldsymbol{u} \boldsymbol{u}^T \boldsymbol{\Sigma} \boldsymbol{u}))
$$

$$
\geq \frac{1}{d-1}(c^2 \lambda_1 - c^2 (\Delta c^2 + \lambda_2)))
$$

$$
= \frac{1}{d-1}(c^2 \Delta - c^4 \Delta) = \frac{\Delta c^2 (1 - c^2)}{d-1}
$$

$$
= \Delta c^2 \mathbb{E}[z^2]
$$

### A.2.3. SELF-TERM BOUND

Since the self-term is monotonically increasing in $X$ and $X \geq 1$ we can take:

$$\mathbb{E}\left[\frac{X^2}{X^2 + Y^2}\middle| c, z\right] \geq \mathbb{E}\left[\frac{1}{1 + Y^2}\middle| c, z\right] \geq \frac{1}{1 + \mathbb{E}[Y^2 | c, z]}$$

where the last inequality applies Jensen's inequality. Note that $f(x) = \frac{1}{1+x}$ is convex for $x > 0$ (since $f''(x) = \frac{2}{(1+x)^3} > 0$), and $Y^2 \geq 0$, so Jensen's inequality gives $\mathbb{E}[f(Y^2)|c, z] \geq f(\mathbb{E}[Y^2|c, z])$. The conditioning on $(c, z)$ fixes the orientations of $\boldsymbol{u}$ and $\boldsymbol{b}$, making $g = \boldsymbol{v}^T \boldsymbol{u}$ and $h = \boldsymbol{v}^T \boldsymbol{b}$ jointly normal with conditional covariance $\mathbb{E}[gh \mid c, z] = \boldsymbol{u}^T \boldsymbol{\Sigma} \boldsymbol{b}$. Since $\boldsymbol{\Sigma}$ is positive semidefinite, Cauchy–Schwarz in the $\boldsymbol{\Sigma}$-inner product gives

$$(\mathbb{E}[gh \mid c, z])^2 = (\boldsymbol{u}^T \boldsymbol{\Sigma} \boldsymbol{b})^2 \leq (\boldsymbol{u}^T \boldsymbol{\Sigma} \boldsymbol{u})(\boldsymbol{b}^T \boldsymbol{\Sigma} \boldsymbol{b}) \leq a^2 b^2,$$

where $a^2 = \Delta c^2 + \lambda_2$ and $b^2 = \Delta z^2 + \lambda_2$ are the conditional variance upper bounds derived in Section A.2.1, valid for any $\boldsymbol{\Sigma} \in \mathcal{G}(\lambda_1, \lambda_2, d)$. The *marginal* expectation $\mathbb{E}_{\boldsymbol{b}}[\mathbb{E}[gh|\boldsymbol{b}]] = 0$ used in the cross-term (Section A.2.2) follows from the symmetry of $\boldsymbol{b}$; the conditional expectation $\mathbb{E}[gh|c, z]$ is generically nonzero and feeds the Isserlis correction below. We evaluate $\mathbb{E}[Y^2|c, z]$ using Isserlis's theorem (jointly Gaussian fourth moment):

$$\begin{aligned}
\mathbb{E}[Y^2 \mid c, z] &= \eta^2(2\mathbb{E}[gh \mid c, z]^2 + \mathbb{E}[g^2 \mid c]\mathbb{E}[h^2 \mid z]) \\
&\leq \eta^2(2a^2b^2 + a^2b^2) \\
&= 3\eta^2 a^2 b^2.
\end{aligned}$$

We then get the self-term bound:

$$\mathbb{E}\left[c^2 \frac{X^2}{X^2 + Y^2}\middle| c\right] \geq \mathbb{E}_z\left[\frac{c^2}{1 + 3\eta^2 a^2 b^2}\right]$$

### A.2.4. CROSS-TERM BOUND

For the cross-term, we have:

$$\frac{XY}{X^2 + Y^2} = \frac{(1 + \eta g^2)\eta g h}{(1 + \eta g^2)^2 + \eta^2 g^2 h^2}$$

To further simplify, notice that:

$$\begin{aligned}
\frac{(1 + \eta g^2)\eta g h}{(1 + \eta g^2)^2 + \eta^2 g^2 h^2} &= \frac{(1 + \eta g^2)gh}{1 + 2\eta g^2 + \eta^2 g^4 + \eta^2 g^2 h^2} \\
&\geq \frac{(1 + \eta g^2)gh}{1 + 2\eta g^2 + \eta^2 g^4 + \eta^2 g^2 h^2 + \eta h^2} \\
&= \frac{(1 + \eta g^2)\eta g h}{(1 + \eta g^2)(1 + \eta g^2 + \eta h^2)} \\
&= \frac{\eta g h}{1 + \eta g^2 + \eta h^2}
\end{aligned}$$

We can then lower bound this simpler quantity. We can use the Taylor series linearization:

$$\frac{1}{1 + x} \geq \frac{1}{1 + a} - \frac{x - a}{(1 + a)^2} = l(x, a) \qquad \forall a \geq 0$$

Denote $\tau = \eta g^2 + \eta h^2$. Multiplying the Taylor inequality by $\eta g h$ (which flips the inequality when $gh < 0$) gives the case-by-case bounds:

$$\frac{\eta g h}{1 + \tau} \geq \frac{\eta g h}{1 + a} - \frac{\eta g h(\tau - a)}{(1 + a)^2} \qquad gh \geq 0, \tag{11}$$

$$\frac{\eta g h}{1 + \tau} \leq \frac{\eta g h}{1 + a} - \frac{\eta g h(\tau - a)}{(1 + a)^2} \qquad gh \leq 0. \tag{12}$$

Splitting the expectation by the sign of $gh$,

$$\mathbb{E}\left[\frac{\eta gh}{1+\tau}\right] = \mathbb{E}\left[\frac{\eta gh}{1+\tau}\mathbb{I}_{gh>0}\right] + \mathbb{E}\left[\frac{\eta gh}{1+\tau}\mathbb{I}_{gh<0}\right],$$

we want to lower bound each piece by $\mathbb{E}[\eta gh \cdot l(\tau, a)]$ restricted to the corresponding region; the $gh \geq 0$ piece satisfies this directly by (11), while the $gh \leq 0$ piece flips by (12). The key step is therefore to show

$$\mathbb{E}\left[\left(\frac{\eta gh}{1+\tau} - \eta gh \cdot l(\tau, a)\right)\mathbb{I}_{gh>0}\right] \geq \mathbb{E}\left[\left(\eta gh \cdot l(\tau, a) - \frac{\eta gh}{1+\tau}\right)\mathbb{I}_{gh<0}\right]. \tag{13}$$

Both sides of (13) are non-negative integrals of the same non-negative function $\varphi(g, h) := |\eta gh| \cdot \left(\frac{1}{1+\tau(g,h)} - l(\tau(g, h), a)\right)$ – non-negative because $1/(1 + \tau) \geq l(\tau, a)$ pointwise, and even in $(g, h)$ because $\tau$ depends only on $g^2, h^2$. Hence the substitution $g \mapsto -g$ on the $gh < 0$ region maps it bijectively to the $gh > 0$ region without changing $\varphi$, giving

$$\text{LHS} - \text{RHS} = \int_{gh>0} \varphi(g, h)\big[f(g, h) - f(-g, h)\big]\, dg\, dh \geq 0,$$

where the last inequality uses the pointwise pdf bound $f(g, h) \geq f(-g, h)$ for $gh \geq 0$ (a standard property of bivariate normals with conditional covariance $\mathbb{E}[gh \mid c, z] = \boldsymbol{u}^T\boldsymbol{\Sigma}\boldsymbol{b} \geq 0$). The opposite-sign case $\boldsymbol{u}^T\boldsymbol{\Sigma}\boldsymbol{b} < 0$ is handled symmetrically with the inequalities in (11)–(12) swapped, giving the same final bound up to a sign convention on $\eta\mathbb{E}[gh \mid c, z] = \eta\,\boldsymbol{u}^T\boldsymbol{\Sigma}\boldsymbol{b}$.

Rearranging (13) gives:

$$\mathbb{E}\left[\frac{\eta gh}{1+\tau}\mathbb{I}_{gh>0} + \frac{\eta gh}{1+\tau}\mathbb{I}_{gh<0}\right] \geq \mathbb{E}\left[\eta gh \cdot l(\tau, a)\mathbb{I}_{gh>0} + \eta gh \cdot l(\tau, a)\mathbb{I}_{gh<0}\right]$$

$$\mathbb{E}\left[\frac{\eta gh}{1+\tau}\right] \geq \mathbb{E}[\eta gh \cdot l(\tau, a)]$$

$$= \frac{\eta\mathbb{E}[gh]}{1+a} - \frac{\eta^2\mathbb{E}[g^3h + gh^3] - a\eta\mathbb{E}[gh]}{(1+a)^2}$$

We can apply Isserlis' theorem since $g$ and $h$ are jointly normal to get:

$$\geq \frac{\eta\mathbb{E}[gh]}{1+a} - \frac{3\eta^2\mathbb{E}[gh](\mathbb{E}[g^2] + \mathbb{E}[h^2]) - a\eta\mathbb{E}[gh]}{(1+a)^2}$$

we can then choose $a = 3\eta(\mathbb{E}[g^2] + \mathbb{E}[h^2])$ – which is the RHS minimizer – to cause cancellation and get:

$$\mathbb{E}\left[\frac{\eta gh}{1 + \eta g^2 + \eta h^2}\right] \geq \frac{\eta\mathbb{E}[gh]}{1 + 3\eta(\mathbb{E}[g^2] + \mathbb{E}[h^2])}$$

which is the desired result. Note that we have assumed $g$ and $h$ are positively correlated which assumes $\mathbb{E}[gh] \geq 0$. If $\mathbb{E}[gh] < 0$ we have the corresponding upper bound

$$\mathbb{E}\left[\frac{\eta gh}{1 + \eta g^2 + \eta h^2}\right] \leq \frac{\eta\mathbb{E}[gh]}{1 + 3\eta(\mathbb{E}[g^2] + \mathbb{E}[h^2])} \tag{14}$$

In both cases, we get the same result since the cross-term, multiplied by $2cz$ and integrated over the exploration direction $\boldsymbol{b}$, is governed by $c\,\mathbb{E}[z\,gh] = \Delta c^2\mathbb{E}[z^2] \geq 0$ (Section A.2.2):

$$\mathbb{E}\left[\frac{2czXY}{X^2 + Y^2}\,\Big|\,c\right] \geq \frac{2\eta c\mathbb{E}[zgh]}{1 + 3\eta(\mathbb{E}[g^2] + \mathbb{E}[h^2])}$$

We can plug in the values from A.2.1 and A.2.2 to get our final bound:

$$\mathbb{E}\left[\frac{2czXY}{X^2 + Y^2}\,\Big|\,c\right] \geq \frac{2\eta\Delta c^2\mathbb{E}[z^2]}{1 + 3\eta(a^2 + b^2)}$$

A.2.5. COMBINING BOUNDS AND CONSTRUCTING STOCHASTIC RECURRENCE

So far, we have put together a lower bound on the cosine alignment of the next iterate given the current iterate. The expected squared cosine alignment has bound:

$$\mathbb{E}[c_{t+1}^2|c] \geq \mathbb{E}\left[c^2 \frac{X^2}{X^2+Y^2} + 2cz\frac{XY}{X^2+Y^2}\bigg|c\right]$$

$$\geq \mathbb{E}_z\left[\frac{c^2}{1+3\eta^2a^2b^2} + \frac{2\eta\Delta c^2z^2}{1+3\eta(a^2+b^2)}\bigg|c\right]$$

$$\mathbb{E}[g^2] \leq a^2 = \Delta c^2 + \lambda_2 \qquad \mathbb{E}[h^2] \leq b^2 = \Delta z^2 + \lambda_2$$

where subscripts $t$ are dropped, and $z^2$ is the squared cosine alignment between the random unit vector and true eigenvector $(\bar{\boldsymbol{u}}^T \boldsymbol{b})^2$. This is complicated, but we can use a few inequalities to simplify:

$$\frac{1}{1+x} \geq 1 - x$$
$$a^2b^2 \leq \lambda_1\Delta c^2z^2 + \lambda_1\lambda_2$$
$$a^2 + b^2 = \Delta(c^2+z^2) + 2\lambda_2$$
$$0 \leq c^2 \leq 1$$
$$0 \leq z^2 \leq 1$$
$$\mathbb{E}[z^2|c] = \frac{1-c^2}{d-1}$$

For the conservative bound on $a^2b^2$, we note that:

$$a^2b^2 = (\Delta c^2 + \lambda_2)(\Delta z^2 + \lambda_2)$$
$$= \Delta^2 c^2z^2 + \lambda_2\Delta(c^2+z^2) + \lambda_2^2$$
$$\leq \Delta^2 c^2z^2 + \lambda_2\Delta + \lambda_2^2 \quad (\text{since } c^2 + z^2 \leq 1)$$
$$= \Delta^2 c^2z^2 + \lambda_2(\Delta + \lambda_2)$$
$$\leq \lambda_1\Delta c^2z^2 + \lambda_1\lambda_2 \quad (\text{since } \Delta \leq \lambda_1 \text{ and } \Delta + \lambda_2 \leq \lambda_1)$$

We can apply these to get (carrying the Cauchy–Schwarz Isserlis bound $3a^2b^2$ from Section A.2.3 through the linearization $\frac{1}{1+x} \geq 1-x$):

$$\mathbb{E}[c_{t+1}^2|c] \geq c^2\mathbb{E}_z\left[1 + 2\eta\Delta z^2 - \eta^2(3a^2b^2 + 6\Delta z^2(a^2+b^2))\right]$$
$$\geq c^2\mathbb{E}_z\left[1 + 2\eta\Delta z^2 - \eta^2(3\lambda_1\Delta c^2z^2 + 3\lambda_1\lambda_2 + 6\Delta z^2(\Delta(c^2+z^2)+2\lambda_2))\right]$$
$$\geq c^2\mathbb{E}_z\left[1 + 2\eta\Delta z^2 - \eta^2(3\lambda_1\lambda_2 + 3\lambda_1\Delta z^2 + 12\Delta^2z^2 + 12\Delta\lambda_2z^2)\right]$$
$$= c^2\mathbb{E}_z\left[1 + 2\eta\Delta z^2 - \eta^2(3\lambda_1\lambda_2 + 15\lambda_1\Delta z^2)\right]$$

where the last equality uses $3\lambda_1 + 12\Delta + 12\lambda_2 = 3\lambda_1 + 12(\lambda_1-\lambda_2) + 12\lambda_2 = 15\lambda_1$. The penultimate line uses $c^2 \leq 1$ and $z^2 \leq 1$ to consolidate the $c^2z^2$ and $z^4$ terms into $z^2$. Next, we will define an auxiliary step size:

$$\hat{\eta} = \frac{\Delta\eta}{d-1} \qquad \eta = \frac{(d-1)\hat{\eta}}{\Delta}$$

$$\mathbb{E}[c_{t+1}^2|c] \geq c^2 + 2\hat{\eta}c^2(1-c^2) - Sc^2\hat{\eta}^2 \tag{15}$$

$$S = \frac{3\lambda_1\lambda_2 d^2}{\Delta^2} + \frac{15d\lambda_1}{\Delta}$$

We will also define the sequence $x_t = 1 - \mathbb{E}[c_t^2]$ which will be our convergence metric. The recurrence gives us:

$$x_{t+1} \leq x_t(1 - 2\hat{\eta}(1-x_t)) + S(1-x_t)\hat{\eta}^2$$
$$\leq x_t(1 - 2\hat{\eta}(1-x_t)) + S\hat{\eta}^2 \tag{16}$$

We will analyze these two recurrences on $c_t^2$ and $x_t$ for the warmup and local convergence phases, respectively.

A.2.6. WARMUP PHASE

Here, we will make an argument that if the initial cosine alignment is $c^2 = \epsilon$, it geometrically grows until $c^2 = 0.5$ while using a constant step size schedule.

We will take the recurrence from Equation 15 which is:

$$\mathbb{E}[c_{t+1}^2 | c] \geq c^2(1 + 2\hat{\eta}(1 - c^2) - S\hat{\eta}^2)$$

For the warm-up analysis, we will pick a constant step size which is conservative enough to ensure monotonicity of $c_t^2$. In particular, we have two conditions:

$$\mathbb{E}[c_{t+1}^2 | c] \geq \begin{cases} c^2 & 0 \leq c^2 \leq 0.5 \\ 0.5 & 0.5 \leq c^2 \leq 1 \end{cases}$$

For the first case, we have that:

$$S\hat{\eta}^2 \leq 2\hat{\eta}(1 - c^2)$$

$$\hat{\eta} \leq \frac{2(1 - c^2)}{S} \leq \frac{1}{S}$$

to satisfy this, we will choose $\hat{\eta} = 1/2S$ to ensure $c_{t+1}^2 > c_t^2$ strictly when $0 \leq c_t \leq 0.5$. For the second case, we want to ensure that we do not regress back to $c^2 < 0.5$ once we reach it. We have:

$$c^2 + 2\hat{\eta}c^2(1 - c^2) - Sc^2\hat{\eta}^2 \geq 0.5$$

we can plug in our choice $\hat{\eta} = 1/2S$ to get:

$$c^2 + \frac{1}{S}c^2(1 - c^2) - \frac{1}{4S}c^2 \geq 0.5$$

Factoring out $\frac{1}{S}$ from the last two terms:

$$c^2 + \frac{1}{S}\left(c^2(1 - c^2) - \frac{c^2}{4}\right) \geq 0.5$$

$$c^2 + \frac{1}{S}\left(c^2 - c^4 - \frac{c^2}{4}\right) \geq 0.5$$

$$c^2 + \frac{1}{S}\left(\frac{3c^2}{4} - c^4\right) \geq 0.5$$

Multiplying through by $S$ and rearranging:

$$Sc^2 + \frac{3c^2}{4} - c^4 \geq \frac{S}{2}$$

$$\frac{1}{S}\left[\left(S + \frac{3}{4}\right)c^2 - c^4\right] \geq \frac{1}{2}$$

At this point, we observe that the function $f(c^2) = \frac{1}{S}\left[\left(S + \frac{3}{4}\right)c^2 - c^4\right]$ is concave in $c^2$ on the interval $[0.5, 1]$, since its second derivative with respect to $c^2$ is negative. By the concavity property, a concave function on a closed interval achieves its minimum at one of the endpoints. Therefore, to prove that $f(c^2) \geq 0.5$ for all $c^2 \in [0.5, 1]$, it suffices to verify the inequality at both endpoints.

For $c^2 = 0.5$:

$$\frac{1}{S}\left[\left(S + \frac{3}{4}\right)\frac{1}{2} - \frac{1}{4}\right] = \frac{1}{S}\left[\frac{S}{2} + \frac{3}{8} - \frac{1}{4}\right]$$

$$= \frac{1}{S}\left[\frac{S}{2} + \frac{1}{8}\right]$$

$$= \frac{1}{2} + \frac{1}{8S} \geq \frac{1}{2}$$

For $c^2 = 1$:

$$\frac{1}{S}\left[S + \frac{3}{4} - 1\right] = \frac{1}{S}\left[S - \frac{1}{4}\right]$$
$$= 1 - \frac{1}{4S} \geq \frac{1}{2}$$

where the last inequality holds whenever $S \geq \frac{1}{2}$, which is always satisfied since $S \geq 15d \geq 15$.

Therefore, by concavity, the inequality $f(c^2) \geq 0.5$ holds for all $c^2 \in [0.5, 1]$. We have ensured that $\mathbb{E}[c_t^2]$ is a monotonically increasing sequence until it reaches $0.5$, at which point it will not regress below $0.5$ as long as the step size is no larger than $\hat{\eta} = 1/2S$.

### A.2.7. WARMUP ITERATIONS

We will consider $\epsilon \leq x_t \leq 0.5$ to be the warmup phase of the algorithm. For this, we can take Equation 15 and consider the worst case improvement in a single iteration:

$$\mathbb{E}[c_{t+1}^2|c] \geq c^2(1 + 2\hat{\eta}(1 - c^2) - S\hat{\eta}^2)$$

$$\geq c^2(1 + \hat{\eta} - S\hat{\eta}^2)$$

From here we can choose a constant learning rate $\hat{\eta} = 1/2S$ which ensures improvement at every iteration in the warmup phase. This also maximizes the coefficient to $c_t^2$.

$$\mathbb{E}[c_{t+1}^2|c] \geq c^2(1 + \frac{1}{2S} - \frac{1}{4S})$$
$$= c^2(1 + \frac{1}{4S})$$

Finally, we can apply telescoping and the initial condition $\mathbb{E}[c_0^2] = \epsilon$ to get:

$$\mathbb{E}\left[c_{t_0}^2\Big|c_0\right] \geq \epsilon(1 + \frac{1}{4S})^{t_0} \geq 0.5$$

Solving for the number of iterations $T$ gets us:

$$t_0 \geq \frac{\log(\frac{0.5}{\epsilon})}{\log(1 + \frac{1}{4S})}$$

Using the fact that $\log(1 + x) \geq x/(1 + x)$ gives us the slightly stricter condition:

$$t_0 \geq (4S + 1)\log(\frac{0.5}{\epsilon})$$

Using the fact that $\epsilon = \mathbb{E}[c_0^2] = 1/d$ when $\boldsymbol{u}_0$ is drawn uniformly on the unit sphere completes the proof.

### A.2.8. LOCAL CONVERGENCE

To establish convergence of the sequence 16, we will first apply the local constraint $0 \leq x_t \leq 0.5$ which simplifies the recurrence into:

$$1 - \mathbb{E}[c_{t+1}^2|c_t] = x_{t+1} \leq x_t(1 - \hat{\eta}_t) + S\hat{\eta}_t^2$$

we need to pick a sequence of step sizes $\hat{\eta}_t$. One first important condition is that we used the fact that $x_1 \leq 0.5$ in the derivation of this local recurrence. If at any point the sequence fails to satisfy this condition, then the bound would become invalid. This means that we need $\eta_1$ small enough to ensure that $x_2 \leq 0.5$ and ideally that the whole sequence stays monotonic.

To ensure monotonicity $x_{t+1} \le x_t$, we substitute the upper bound from above and derive a sufficient condition on the step size. We require:

$$
\begin{aligned}
x_{t+1} &\le x_t && \text{(desired monotonicity)} \\
x_t(1 - \hat{\eta}_t) + S\hat{\eta}_t^2 &\le x_t && \text{(substituting upper bound for } x_{t+1}) \\
S\hat{\eta}_t^2 &\le \hat{\eta}_t x_t && \text{(rearranging)} \\
\hat{\eta}_t &\le \frac{x_t}{S} && \text{(dividing by } \hat{\eta}_t > 0)
\end{aligned}
$$

Plugging in the initial condition $x_1 = 0.5$, we get that:

$$
\hat{\eta}_1 \le \frac{1}{2S}
$$

We will choose the step size sequence:

$$
\hat{\eta}_t = \frac{K}{T + t}
$$

and choose suitable $K$ and $T$ which will satisfy this condition.

A.2.9. UNROLLING THE RECURRENCE

Consider the recurrence:

$$
x_{t+1} \le x_t(1 - \hat{\eta}_t) + S\hat{\eta}_t^2
$$

The recurrence relation can be unrolled to give the following closed form:

$$
x_{t+1} \le \underbrace{x_1 \prod_{i=1}^{t}(1 - \hat{\eta}_i)}_{\text{first summand}} + \underbrace{\sum_{j=1}^{t} S\hat{\eta}_j^2 \prod_{i=j+1}^{t}(1 - \hat{\eta}_i)}_{\text{second summand}}
$$

To bound the first summand, applying $1 + x \le \exp(x)$ gives us:

$$
\begin{aligned}
\prod_{i=1}^{t}(1 - \hat{\eta}_i) &\le \exp\left(-\sum_{i=1}^{t}\hat{\eta}_i\right) \\
&\le \exp\left(-K\ln\left(\frac{T + t + 1}{T + 1}\right)\right) \\
&= \left(\frac{T + 1}{T + t + 1}\right)^K
\end{aligned}
$$

To bound the second summand, we will need the following auxiliary bounds (proofs in A.3.1 and A.3.2):

$$
\sum_{i=j+1}^{t}\hat{\eta}_i = \sum_{i=j+1}^{t}\frac{K}{T + i} \ge K\ln\left(\frac{T + t + 1}{T + j + 1}\right)
$$

$$
\sum_{i=j+1}^{t}\hat{\eta}_i^2 = \sum_{i=j+1}^{t}\frac{K^2}{(T + i)^2} \le \frac{K^2}{T + j}
$$

Applying these bounds to the second summand gives us:

$$\sum_{j=1}^{t} S\hat{\eta}_j^2 \prod_{i=j+1}^{t} (1 - \hat{\eta}_i) \leq SK^2 \sum_{j=1}^{t} \frac{1}{(T+j)^2} \left[ \left( \frac{T+j+1}{T+t+1} \right)^K \right]$$

$$\leq \frac{SK^2}{(T+t+1)^K} \sum_{j=1}^{t} \frac{(T+j+1)^K}{(T+j)^2}$$

$$\leq \frac{SK^2}{(T+t+1)^K} \sum_{j=1}^{t} \frac{(T+j+1)^2}{(T+j)^2} (T+j+1)^{K-2}$$

using the fact that the expression is monotonically decreasing in $j$, we get:

$$\frac{(T+j+1)^2}{(T+j)^2} \leq \frac{(T+2)^2}{(T+1)^2}$$

we get

$$\sum_{j=1}^{t} S\hat{\eta}_j^2 \prod_{i=j+1}^{t} (1 - \hat{\eta}_i) \leq \frac{SK^2}{(T+t+1)^K} \left( \frac{T+2}{T+1} \right)^2 \sum_{j=1}^{t} (T+j+1)^{K-2}$$

If we take $K \geq 2$ we can bound the sum as:

$$\sum_{j=1}^{t} (T+j+1)^{K-2} \leq \int_{1}^{t+1} (T+x)^{K-2} dx$$

$$= \frac{1}{K-1} ((T+t+1)^{K-1} - (T+1)^{K-1})$$

$$\leq \frac{(T+t+1)^{K-1}}{K-1}$$

Finishing the bound:

$$\sum_{j=1}^{t} S\hat{\eta}_j^2 \prod_{i=j+1}^{t} (1 - \hat{\eta}_i) \leq \frac{SK^2}{(T+t+1)^K} \left( \frac{T+2}{T+1} \right)^2 \sum_{j=1}^{t} (T+j+1)^{K-2}$$

$$\leq \frac{SK^2}{K-1} \left( \frac{T+2}{T+1} \right)^2 \frac{(T+t+1)^{K-1}}{(T+t+1)^K}$$

$$= \frac{SK^2}{K-1} \left( \frac{T+2}{T+1} \right)^2 \left( \frac{1}{T+t+1} \right)$$

Then combining with the other part we get our final bound:

$$x_{t+1} \leq \frac{C_1}{T+t+1} + \frac{x_1 C_2}{(T+t+1)^K}$$

with constants:

$$C_1 = \frac{SK^2}{K-1} \left( \frac{T+2}{T+1} \right)^2 \qquad C_2 = (T+1)^K$$

Now we recall that $K$ and $T$ are constants chosen to set the step size schedule, as long as we satisfy the constraints:

$$K \geq 2 \qquad \hat{\eta}_1 = \frac{K}{T+1} \leq \frac{1}{2S}$$

where $S$ is related to the problem parameters. If we fix $K = 2$ we get:

$$T \geq 4S$$

So we will pick $T = 4S$. We can then compute:

$$C_1 = 4S \left(\frac{4S + 2}{4S + 1}\right)^2 \leq 4S + 2 \qquad C_2 = (4S + 1)^2$$

which completes the proof. We combine the coefficient $C_2$ with the initial condition $x_1 = 0.5$ and count from iterations starting at $t_0$ to get Theorem 4.1.

### A.2.10. FIXED POINT WITH CONSTANT STEP SIZE

Here we will compute a simple steady state tracking result using the warmup recurrence 15. We begin with:

$$\mathbb{E}[c_{t+1}^2 | c] \geq c^2 + 2\hat{\eta}c^2(1 - c^2) - Sc^2\hat{\eta}^2$$

We will try to find a steady state when we consider a fixed learning rate. Solving for $c^2$ gives us:

$$2\hat{\eta}c^2(1 - c^2) = Sc^2\hat{\eta}^2$$
$$c^2 = 1 - \frac{S\hat{\eta}}{2}$$

or in terms of the sine alignment:

$$x^* = \frac{S\hat{\eta}}{2}$$

So for instance if we pick a choice of step size $\hat{\eta} = 1/2S$ as in the warmup phase, we get a fixed point at $x = 0.25$. This is a stable fixed point since the recurrence has a first-order difference which is positive on $[0, x)$ and negative on $(x, 1]$

### A.2.11. TRACKING A CHANGING EIGENVECTOR

Let's assume now that the leading eigenvector $\bar{u}$ of $\Sigma$ is allowed to drift over time. We will not impose any particular structure on the motion, but we will restrict it so that:

$$(\bar{u}_t^T \bar{u}_{t+1})^2 \geq 1 - V$$

for some maximum "velocity" $V$. We define the alignment with the new and old eigenvectors:

$$c_{t+1} = \bar{u}_{t+1}^T u_{t+1} \qquad \tilde{c}_{t+1} = \bar{u}_t^T u_{t+1}$$

To relate these, we decompose the new eigenvector:

$$\bar{u}_{t+1} = (\bar{u}_t^T \bar{u}_{t+1})\bar{u}_t + \sqrt{1 - (\bar{u}_t^T \bar{u}_{t+1})^2}\, \bar{w}$$

where $\bar{w}$ is a unit vector orthogonal to $\bar{u}_t$ representing the drift direction. Similarly:

$$u_{t+1} = \tilde{c}_{t+1}\bar{u}_t + \sqrt{1 - \tilde{c}_{t+1}^2}\, w'$$

where $w'$ is orthogonal to $\bar{u}_t$. Then:

$$c_{t+1} = \bar{u}_{t+1}^T u_{t+1} = (\bar{u}_t^T \bar{u}_{t+1})\tilde{c}_{t+1} + \sqrt{V}\sqrt{1 - \tilde{c}_{t+1}^2}\,(\bar{w}^T w')$$

Squaring:

$$c_{t+1}^2 = (1 - V)\tilde{c}_{t+1}^2 + V(1 - \tilde{c}_{t+1}^2)(\bar{w}^T w')^2 + 2\sqrt{V(1 - V)}\,\tilde{c}_{t+1}\sqrt{1 - \tilde{c}_{t+1}^2}\,(\bar{w}^T w')$$

The direction $w'$ depends on the algorithm's exploration via the random orthogonal direction $b_t$, while $\bar{w}$ is the eigenvector drift direction. Assuming these are independent and that $\bar{w}^T w'$ has zero mean (the drift direction is not systematically aligned with or against the algorithm's exploration), we have $\mathbb{E}[\bar{w}^T w' | \tilde{c}_{t+1}] = 0$. Taking expectations:

$$\mathbb{E}[c_{t+1}^2 | \tilde{c}_{t+1}] = (1 - V)\tilde{c}_{t+1}^2 + V(1 - \tilde{c}_{t+1}^2)\mathbb{E}[(\bar{w}^T w')^2]$$
$$\geq (1 - V)\tilde{c}_{t+1}^2$$

Combined with the stationary analysis bound $\mathbb{E}[\tilde{c}_{t+1}^2 | c] \geq c^2 + 2\hat{\eta}c^2(1 - c^2) - Sc^2\hat{\eta}^2$, we obtain:

$$\mathbb{E}[c_{t+1}^2 | c] \geq (1 - V)\mathbb{E}[\tilde{c}_{t+1}^2 | c]$$
$$\geq (1 - V)(c^2 + 2\hat{\eta}c^2(1 - c^2) - Sc^2\hat{\eta}^2)$$
$$\geq c^2 + 2\hat{\eta}c^2(1 - c^2) - Sc^2\hat{\eta}^2 - Vc^2 - 2V\hat{\eta}c^2$$
$$= c^2 + 2\hat{\eta}c^2(1 - c^2) - \tilde{S}c^2\hat{\eta}^2$$

where:

$$\tilde{S} = S + \frac{V}{\hat{\eta}^2} + \frac{2V}{\hat{\eta}}$$

We can use our fixed point theorem to optimize for the best step size:

$$x^* = \frac{\tilde{S}\hat{\eta}}{2}$$
$$= \frac{\hat{\eta}S + 2V + \frac{V}{\hat{\eta}}}{2}$$

We can find the minimizer by setting the derivative with respect to $\hat{\eta}$ equal to zero:

$$S - \frac{V}{\hat{\eta}^2} = 0$$

$$\hat{\eta} = \sqrt{\frac{V}{S}}$$

This lets us compute the fixed point:

$$x^* = V + \sqrt{VS}$$

### A.3. Auxiliary Lemmas

#### A.3.1. HARMONIC SUM

Let $S = \sum_{i=j+1}^{t} \frac{1}{T+i}$. Since the function $f(x) = \frac{1}{T+x}$ is strictly decreasing on the interval $x \in [j+1, t]$, we can bound the sum from below using an integral comparison:

$$S = \sum_{i=j+1}^{t} \frac{1}{T+i} \geq \int_{j+1}^{t+1} \frac{1}{T+x} dx$$

Evaluating the integral:

$$\int_{j+1}^{t+1} \frac{1}{T+x} dx = \ln(T+x)\Big|_{j+1}^{t+1}$$
$$= \ln(T+t+1) - \ln(T+j+1)$$
$$= \ln\left(\frac{T+t+1}{T+j+1}\right)$$

Therefore:

$$\sum_{i=j+1}^{t} \frac{1}{T+i} \geq \ln\left(\frac{T+t+1}{T+j+1}\right)$$

### A.3.2. Squared Harmonic Sum

Let $S = \sum_{i=j+1}^{t} \frac{1}{(T+i)^2}$. Since $f(x) = \frac{1}{(T+x)^2}$ is strictly decreasing for positive $x$, we can bound it using the integral:

$$
\begin{aligned}
\sum_{i=j+1}^{t} \frac{1}{(T+i)^2} &\leq \int_{j}^{t} \frac{1}{(T+x)^2} dx \\
&= -\frac{1}{T+x}\Big|_{j}^{t} \\
&= \frac{1}{T+j} - \frac{1}{T+t} \\
&\leq \frac{1}{T+j}
\end{aligned}
$$

# B. Information-Theoretic Lower Bounds for Compressed Sensing

This appendix establishes information-theoretic lower bounds for estimating the principal eigenvector using compressed measurements. For Gaussian data with two linear measurements per iteration, we prove a minimax risk lower bound of $\Omega(\lambda_1 \lambda_2 d^2 / (\Delta^2 t))$ in the *adaptive* setting — matching our upper bound and establishing the optimality of the $d^2$ scaling — and a stronger $\Omega(\lambda_2^2 d^3 / (\Delta^2 t))$ bound in the *non-adaptive* setting, reflecting the additional cost of fixing the measurement scheme before observing data.

## B.1. Problem Setup

Consider $t$ i.i.d. samples $\boldsymbol{v}_1, \ldots, \boldsymbol{v}_t \in \mathbb{R}^d$ drawn from $\mathcal{N}(\boldsymbol{0}, \boldsymbol{\Sigma})$. To prove a lower bound, we construct a family of covariance matrices of the form

$$\boldsymbol{\Sigma} = \lambda_2 \mathbf{I} + \Delta \bar{\boldsymbol{u}} \bar{\boldsymbol{u}}^T$$

where $\bar{\boldsymbol{u}}$ is the principal eigenvector, $\lambda_1 = \lambda_2 + \Delta$ is the leading eigenvalue, and all other eigenvalues equal $\lambda_2$. Each such covariance lies inside $\mathcal{G}(\lambda_1, \lambda_2, d)$ and represents the hardest case for estimation: when all trailing eigenvalues are equal, the noise subspace provides no additional structure to exploit.

The problem of estimating the principal eigenvector is *rotationally invariant*: for any orthogonal matrix $\boldsymbol{Q}$, if $\boldsymbol{v} \sim \mathcal{N}(\boldsymbol{0}, \boldsymbol{\Sigma})$, then $\boldsymbol{Q}\boldsymbol{v} \sim \mathcal{N}(\boldsymbol{0}, \boldsymbol{Q}\boldsymbol{\Sigma}\boldsymbol{Q}^T)$, and the principal eigenvector transforms as $\bar{\boldsymbol{u}} \mapsto \boldsymbol{Q}\bar{\boldsymbol{u}}$. Since the squared distance $\rho^2(\hat{\boldsymbol{u}}, \bar{\boldsymbol{u}}) = 1 - |\langle \hat{\boldsymbol{u}}, \bar{\boldsymbol{u}} \rangle|^2$ is invariant under orthogonal transformations, the minimax risk is unchanged by rotations. Consequently, the information-theoretic difficulty of estimation depends *only* on the spectral gap $\Delta = \lambda_1 - \lambda_2$ and the eigenvalue structure — not on the specific orientation of the principal subspace or the choice of basis for $\mathbb{R}^d$.

Our lower bound construction considers a family of such covariances, each with a different principal eigenvector $\bar{\boldsymbol{u}}_v$ indexed by vertices $v$ of a hypercube. By the rotational invariance above, we may without loss of generality work in a coordinate system where the hypercube is centered at the standard basis vector $\boldsymbol{e}_1$, so that the perturbed eigenvectors $\bar{\boldsymbol{u}}_v$ are small perturbations of $\boldsymbol{e}_1$. Note that the individual covariance matrices $\boldsymbol{\Sigma}_v = \lambda_2 \mathbf{I} + \Delta \bar{\boldsymbol{u}}_v \bar{\boldsymbol{u}}_v^T$ are *not* diagonal — only when $\bar{\boldsymbol{u}}_v = \boldsymbol{e}_1$ exactly would the covariance be diagonal.

Our goal is to show that estimating the principal eigenvector $\bar{\boldsymbol{u}}$ from this family of covariances requires $\Omega(d^2)$ samples when using only two adaptive measurements per iteration.

We consider two measurement paradigms: (1) full-dimensional observations where each sample $\boldsymbol{v}_i$ is observed directly, and (2) adaptive compressed measurements using two linear measurements per sample with unit vectors $\boldsymbol{a}_i$ and $\boldsymbol{b}_i$ chosen adaptively based on all prior observations.

In the compressed setting, at each iteration $i$ we only observe two scalar values $[\boldsymbol{a}_i^T \boldsymbol{v}_i, \boldsymbol{b}_i^T \boldsymbol{v}_i]$.

### B.1.1. Key Proof Technique: Averaged Assouad's Lemma

Assouad's lemma extends Le Cam's two-point hypothesis test to multiple hypotheses indexed by vertices of an $s$-dimensional hypercube:

$$\mathcal{F}_s = \{P_v : v \in V = \{-1, 1\}^s\}$$

We have $2^s$ probability distributions to distinguish, each associated with a true eigenvector $\bar{\boldsymbol{u}}_v$ and distribution $P_v$. For $v \in V$ and $j \in \{1, \ldots, s\}$, let $v^{(j)}$ denote the vertex obtained by flipping the $j$-th coordinate of $v$.

**Lemma B.1** (Averaged Assouad's Lemma). *For any estimator $\hat{\boldsymbol{u}}_t$ based on $t$ samples and squared distance $\rho^2(\cdot, \cdot)$:*

$$\inf_{\hat{\boldsymbol{u}}_t} \max_{v \in V} \mathbb{E}_{P_v} \rho(\hat{\boldsymbol{u}}_t, \bar{\boldsymbol{u}}_v)^2 \geq \frac{s}{8} \cdot \min_{H(v,v') \geq 1} \frac{\rho^2(\bar{\boldsymbol{u}}_v, \bar{\boldsymbol{u}}_{v'})}{H(v, v')} \cdot \left(1 - \frac{1}{s} \sum_{j=1}^{s} \frac{1}{2^{s-1}} \sum_{\substack{v \in V \\ v[j]=1}} TV\left(P_v^{(t)}, P_{v^{(j)}}^{(t)}\right)\right)$$

*where $H(v, v')$ denotes the Hamming distance between $v$ and $v'$, and the double sum averages the total variation distance over all $s \cdot 2^{s-1}$ adjacent pairs.*

*Proof.* This is a strengthening of the standard (min-form) Assouad's lemma (see, e.g., van der Vaart (Van der Vaart, 1998), Lemma 24.3), obtained by retaining the per-coordinate average rather than collapsing to the worst-case affinity. The averaged form is stated as Lemma 8.5.2 of Duchi (Duchi, 2023); we give the derivation below.

*Step 1 (Max ≥ average, triangle inequality).* Let $\hat{v} = \operatorname{argmin}_{v' \in V} \rho(\hat{u}_t, \bar{u}_{v'})$. Since $\rho(u, v) = |\sin \angle(u, v)|$ satisfies the triangle inequality on projective space, $\rho(\bar{u}_{\hat{v}}, \bar{u}_v) \le 2\rho(\hat{u}_t, \bar{u}_v)$, giving $\rho^2(\hat{u}_t, \bar{u}_v) \ge \frac{1}{4} \min_{H \ge 1}(\rho^2/H) \cdot H(\hat{v}, v)$. Combining $\max_v \ge 2^{-s} \sum_v$ with $H(\hat{v}, v) = \sum_j \mathbf{1}[\hat{v}[j] \ne v[j]]$:

$$\max_v \mathbb{E}_{P_v} \rho^2 \ge \frac{1}{4} \min_{H \ge 1} \frac{\rho^2}{H} \cdot \sum_{j=1}^{s} \frac{1}{2^s} \sum_{v \in V} P_v(\hat{v}[j] \ne v[j]).$$

*Step 2 (Le Cam per coordinate).* For each $j$, group vertices into pairs $(v, v^{(j)})$ differing in coordinate $j$ only. By Le Cam's two-point lemma, $P_v(\hat{v}[j] \ne 1) + P_{v^{(j)}}(\hat{v}[j] \ne -1) \ge 1 - \mathrm{TV}\left(P_v^{(t)}, P_{v^{(j)}}^{(t)}\right)$. Averaging the $2^{s-1}$ pairs (vertices with $v[j] = 1$) yields

$$\frac{1}{2^s} \sum_{v \in V} P_v(\hat{v}[j] \ne v[j]) \ge \frac{1}{2}\left(1 - \frac{1}{2^{s-1}} \sum_{v:\, v[j]=1} \mathrm{TV}\left(P_v^{(t)}, P_{v^{(j)}}^{(t)}\right)\right).$$

*Step 3 (Combine).* Summing over $j$ and factoring out $s$ converts the bracket into $1 - (1/s) \sum_j (1/2^{s-1}) \sum_{v:v[j]=1} \mathrm{TV}$; the prefactor becomes $(1/4) \cdot (1/2) \cdot s = s/8$, yielding the lemma. $\qquad\square$

The first minimum is over *all* pairs of distinct vertices (normalized by Hamming distance), while the affinity factor uses the *average* total variation across adjacent pairs. The averaged form is essential when measurements may be adaptive: a strategy can concentrate measurement energy on a subset of coordinates, distinguishing those adjacent pairs sharply at the cost of others. The min-form bound would then be dominated by the easy pairs and become vacuous, whereas the averaged form aggregates per-coordinate hardness across the global energy budget.

### B.1.2. HYPERCUBE CONSTRUCTION

We define the true eigenvectors $\bar{u}_v$ as perturbations of the standard unit vector $e_1$:

$$\bar{u}_v = e_1 \sqrt{1 - \frac{s\beta^2}{4}} + \frac{\beta}{2} \sum_{i=1}^{s} v[i] e_{i+1}$$

where $s \le d - 1$ and $\beta^2 \le 2/s$ ensures perturbations are less than $90°$ apart.

By construction, $\|\bar{u}_v\| = 1$. When $v$ and $v'$ differ in one coordinate, the distance is $\|\bar{u}_v - \bar{u}_{v'}\| = \beta$.

For the sine-squared distance metric $\rho^2(u, v) = 1 - (u^T v)^2$ and adjacent vertices (Hamming distance 1):

$$\alpha^2 := \min_{v,v' \in \mathrm{Adj}(V)} \rho^2(\bar{u}_v, \bar{u}_{v'}) = \min_{v,v' \in \mathrm{Adj}(V)} 1 - (\bar{u}_v^T \bar{u}_{v'})^2$$

$$\alpha^2 = 1 - \left(1 - \frac{\beta^2}{2}\right)^2 = \beta^2 - \frac{\beta^4}{4}$$

Note that $\beta^2 \le 2\alpha^2$ for valid $\beta$. To see this, write $\alpha^2 = \beta^2(1 - \beta^2/4)$. Since $\beta^2 \le 2/s \le 2$, we have $1 - \beta^2/4 \ge 1/2$, so $\alpha^2 \ge \beta^2/2$, which rearranges to $\beta^2 \le 2\alpha^2$.

For Assouad's Lemma, we must compute $\min_{H(v,v') \ge 1} \frac{\rho^2(\bar{u}_v, \bar{u}_{v'})}{H(v,v')}$. For vertices with Hamming distance $k$, we have:

$$\bar{u}_v^T \bar{u}_{v'} = 1 - \frac{s\beta^2}{4} + \frac{\beta^2}{4}(s - 2k) = 1 - \frac{k\beta^2}{2}$$

Thus:

$$\rho^2(\bar{u}_v, \bar{u}_{v'}) = 1 - \left(1 - \frac{k\beta^2}{2}\right)^2 = k\beta^2 - \frac{k^2\beta^4}{4}$$

The ratio is:

$$\frac{\rho^2(\bar{u}_v, \bar{u}_{v'})}{H(v, v')} = \frac{k\beta^2 - \frac{k^2\beta^4}{4}}{k} = \beta^2 - \frac{k\beta^4}{4} = \beta^2\left(1 - \frac{k\beta^2}{4}\right)$$

This is minimized at $k = s$ (antipodal vertices), giving:

$$\min_{H(v,v') \geq 1} \frac{\rho^2(\bar{\boldsymbol{u}}_v, \bar{\boldsymbol{u}}_{v'})}{H(v,v')} = \beta^2 \left(1 - \frac{s\beta^2}{4}\right)$$

Since we require $\beta^2 \leq \frac{2}{s}$ for construction validity, we have $\frac{s\beta^2}{4} \leq \frac{1}{2}$, and therefore:

$$\min_{H(v,v') \geq 1} \frac{\rho^2(\bar{\boldsymbol{u}}_v, \bar{\boldsymbol{u}}_{v'})}{H(v,v')} \geq \frac{\beta^2}{2} \geq \frac{\alpha^2}{2}$$

where the last inequality uses $\alpha^2 = \beta^2 - \frac{\beta^4}{4} \leq \beta^2$.

For each unit vector $\bar{\boldsymbol{u}}_v$, we associate a covariance matrix and multivariate normal distribution:

$$\boldsymbol{\Sigma}_v = \lambda_2 \mathbf{I} + \Delta \bar{\boldsymbol{u}}_v \bar{\boldsymbol{u}}_v^T, \qquad P_v = \mathcal{N}(\mathbf{0}, \boldsymbol{\Sigma}_v)$$

where $\Delta = \lambda_1 - \lambda_2$ is the eigengap.

### B.1.3. FULL-DIMENSIONAL CASE (BASELINE)

For reference, we first establish the lower bound for fully-observed data. The KL divergence between two zero-mean Gaussians with covariances $\boldsymbol{\Sigma}_v$ and $\boldsymbol{\Sigma}_{v'}$ is:

$$D_{KL}(P_v, P_{v'}) = \frac{1}{2}\left(\text{tr}(\boldsymbol{\Sigma}_{v'}^{-1}\boldsymbol{\Sigma}_v) + \ln\frac{|\boldsymbol{\Sigma}_{v'}|}{|\boldsymbol{\Sigma}_v|} - d\right)$$

Since both covariance matrices have the form $\boldsymbol{\Sigma}_v = \lambda_2\mathbf{I} + \Delta\bar{\boldsymbol{u}}_v\bar{\boldsymbol{u}}_v^T$, their determinants are identical: $|\boldsymbol{\Sigma}_v| = |\boldsymbol{\Sigma}_{v'}| = \lambda_1\lambda_2^{d-1}$, so the log-determinant term vanishes. We focus on the trace term.

Using the Woodbury identity (Sherman-Morrison formula) on $\boldsymbol{\Sigma}_{v'}^{-1}$:

$$\boldsymbol{\Sigma}_{v'}^{-1} = (\lambda_2\mathbf{I} + \Delta\bar{\boldsymbol{u}}_{v'}\bar{\boldsymbol{u}}_{v'}^T)^{-1} = \frac{1}{\lambda_2}\left(\mathbf{I} - \frac{\Delta\bar{\boldsymbol{u}}_{v'}\bar{\boldsymbol{u}}_{v'}^T}{\lambda_2 + \Delta}\right) = \frac{1}{\lambda_2}\left(\mathbf{I} - \frac{\Delta\bar{\boldsymbol{u}}_{v'}\bar{\boldsymbol{u}}_{v'}^T}{\lambda_1}\right)$$

Computing the trace:

$$\begin{aligned}
\text{tr}(\boldsymbol{\Sigma}_{v'}^{-1}\boldsymbol{\Sigma}_v) &= \frac{1}{\lambda_2}\text{tr}\left[\left(\mathbf{I} - \frac{\Delta\bar{\boldsymbol{u}}_{v'}\bar{\boldsymbol{u}}_{v'}^T}{\lambda_1}\right)\left(\lambda_2\mathbf{I} + \Delta\bar{\boldsymbol{u}}_v\bar{\boldsymbol{u}}_v^T\right)\right] \\
&= \frac{1}{\lambda_2}\text{tr}\left(\lambda_2\mathbf{I} + \Delta\bar{\boldsymbol{u}}_v\bar{\boldsymbol{u}}_v^T - \frac{\lambda_2\Delta\bar{\boldsymbol{u}}_{v'}\bar{\boldsymbol{u}}_{v'}^T}{\lambda_1} - \frac{\Delta^2\bar{\boldsymbol{u}}_{v'}\bar{\boldsymbol{u}}_{v'}^T\bar{\boldsymbol{u}}_v\bar{\boldsymbol{u}}_v^T}{\lambda_1}\right) \\
&= d + \frac{\Delta}{\lambda_2} - \frac{\Delta}{\lambda_1} - \frac{\Delta^2(\bar{\boldsymbol{u}}_{v'}^T\bar{\boldsymbol{u}}_v)^2}{\lambda_1\lambda_2} \\
&= d + \frac{\Delta}{\lambda_2}\left(1 - \frac{\lambda_2}{\lambda_1} - \frac{\Delta(\bar{\boldsymbol{u}}_{v'}^T\bar{\boldsymbol{u}}_v)^2}{\lambda_1}\right) \\
&= d + \frac{\Delta}{\lambda_2} \cdot \frac{\Delta(1 - (\bar{\boldsymbol{u}}_{v'}^T\bar{\boldsymbol{u}}_v)^2)}{\lambda_1} = d + \frac{\Delta^2\alpha^2}{\lambda_1\lambda_2}
\end{aligned}$$

where we used $1 - (\bar{\boldsymbol{u}}_{v'}^T\bar{\boldsymbol{u}}_v)^2 = \alpha^2$ (the sine-squared distance between adjacent vertices). Therefore:

$$D_{KL}(P_v, P_{v'}) = \frac{1}{2}\left(\frac{\Delta^2\alpha^2}{\lambda_1\lambda_2}\right) = \frac{\Delta^2\alpha^2}{2\lambda_1\lambda_2}$$

Over $t$ i.i.d. samples, KL divergence accumulates linearly:

$$D_{KL}(P_v^{(t)}, P_{v'}^{(t)}) = \frac{t\Delta^2\alpha^2}{2\lambda_1\lambda_2}$$

Applying Pinsker's inequality:

$$\text{TV}(P_v^{(t)}, P_{v'}^{(t)}) \leq \sqrt{\frac{1}{2} D_{KL}(P_v^{(t)}, P_{v'}^{(t)})} = \frac{\Delta\alpha}{2}\sqrt{\frac{t}{\lambda_1\lambda_2}}$$

Since the per-pair KL divergence above depends only on $\alpha^2$ (which is the same for every adjacent pair by construction), the averaged total variation in Lemma B.1 equals the per-pair total variation. Applying Lemma B.1 with $s = d-1$ and the Hamming distance bound $\min_{H\geq 1} \rho^2/H \geq \alpha^2/2$:

$$\inf_{\hat{\boldsymbol{u}}_t} \max_{v\in V} \mathbb{E}_{P_v} \rho(\hat{\boldsymbol{u}}_t, \bar{\boldsymbol{u}}_v)^2 \geq \frac{s}{8} \cdot \frac{\alpha^2}{2} \cdot \left(1 - \frac{\Delta\alpha}{2}\sqrt{\frac{t}{\lambda_1\lambda_2}}\right) = \frac{s\alpha^2}{16}\left(1 - \frac{\Delta\alpha}{2}\sqrt{\frac{t}{\lambda_1\lambda_2}}\right)$$

We choose $\alpha = \frac{1}{\Delta}\sqrt{\frac{\lambda_1\lambda_2}{t}}$, which gives:

$$\alpha^2 = \frac{\lambda_1\lambda_2}{\Delta^2 t}, \quad \frac{\Delta\alpha}{2}\sqrt{\frac{t}{\lambda_1\lambda_2}} = \frac{1}{2}$$

For the construction to be valid, we require $\beta^2 \leq 2/s$. Since $\alpha^2 = \beta^2 - \beta^4/4$, we have $\beta^2 = 2 - 2\sqrt{1-\alpha^2}$ (taking the smaller root).

This imposes:

$$\beta^2 = 2 - 2\sqrt{1 - \frac{\lambda_1\lambda_2}{\Delta^2 t}} \leq \frac{2}{s}$$

$$1 - \frac{1}{s} \leq \sqrt{1 - \frac{\lambda_1\lambda_2}{\Delta^2 t}}$$

$$\frac{\lambda_1\lambda_2}{\Delta^2 t} \leq \frac{2s-1}{s^2}$$

With $s = d-1$:

$$t \geq \frac{(d-1)^2\lambda_1\lambda_2}{\Delta^2(2d-3)}$$

$$\geq \frac{(d-1)^2\lambda_1\lambda_2}{\Delta^2(2d-2)}$$

$$= \frac{(d-1)\lambda_1\lambda_2}{2\Delta^2}$$

This defines the valid number of samples required for the bound to hold. Substituting $\alpha$ into Assouad's Lemma:

$$\text{Lower Bound} = \frac{(d-1)}{16} \cdot \frac{\lambda_1\lambda_2}{\Delta^2 t} \cdot \frac{1}{2} = \frac{(d-1)\lambda_1\lambda_2}{32\Delta^2 t}$$

Therefore:

$$\boxed{\inf_{\hat{\boldsymbol{u}}_t} \max_{v\in V} \mathbb{E}_{P_v} \rho(\hat{\boldsymbol{u}}_t, \bar{\boldsymbol{u}}_v)^2 \geq \frac{(d-1)\lambda_1\lambda_2}{32\Delta^2 t}} \tag{17}$$

valid for $t \geq \frac{(d-1)\lambda_1\lambda_2}{2\Delta^2}$. This establishes the minimax lower bound for full-dimensional PCA, which is $\Theta(d\lambda_1\lambda_2/\Delta^2 t)$.

## B.2. Adaptive Compressed Measurements Lower Bound

We now extend to the compressed measurement setting where at each iteration $i \in \{1, \ldots, t\}$ we observe:

$$\boldsymbol{x}_i = \begin{bmatrix} \boldsymbol{a}_i^T \\ \boldsymbol{b}_i^T \end{bmatrix} \boldsymbol{v}_i = \boldsymbol{A}_i \boldsymbol{v}_i$$

where $\boldsymbol{a}_i, \boldsymbol{b}_i \in \mathbb{R}^d$ are unit vectors chosen adaptively based on all prior observations. The key question is: how much information do two scalar measurements provide toward distinguishing hypotheses?

B.2.1. REDUCTION TO ORTHONORMAL MEASUREMENTS

Without loss of generality, we may assume each $\boldsymbol{A}_i$ has orthonormal rows. The key insight is that the information content in the measurements $\boldsymbol{x}_i = \boldsymbol{A}_i \boldsymbol{v}_i$ depends only on the 2-dimensional subspace spanned by the rows of $\boldsymbol{A}_i$, not on the specific choice of basis vectors within that subspace.

Since any two linearly independent measurement vectors span a 2-dimensional subspace that admits an orthonormal basis, we can work with orthonormal rows without restricting the class of admissible measurement strategies.

Formally, consider the QR decomposition $\boldsymbol{A}_i^T = \boldsymbol{Q}_i \boldsymbol{R}_i$ where $\boldsymbol{Q}_i \in \mathbb{R}^{d \times 2}$ has orthonormal columns ($\boldsymbol{Q}_i^T \boldsymbol{Q}_i = \boldsymbol{I}_2$) and $\boldsymbol{R}_i \in \mathbb{R}^{2 \times 2}$ is invertible (since $\boldsymbol{A}_i$ has full row rank). The KL divergence between compressed observations is:

$$D_{KL}(P_{x,v}, P_{x,v'}) = \frac{1}{2}\left(\operatorname{tr}(\boldsymbol{\Sigma}_{x,v'}^{-1}\boldsymbol{\Sigma}_{x,v}) + \ln\frac{|\boldsymbol{\Sigma}_{x,v'}|}{|\boldsymbol{\Sigma}_{x,v}|} - 2\right)$$

where $\boldsymbol{\Sigma}_{x,v} = \boldsymbol{A}_i \boldsymbol{\Sigma}_v \boldsymbol{A}_i^T$.

We verify that the KL divergence is invariant to replacing $\boldsymbol{A}_i$ with $\boldsymbol{Q}_i^T$:

For the determinant term, using $\boldsymbol{A}_i = \boldsymbol{R}_i^T \boldsymbol{Q}_i^T$ and the determinant product rule:

$$\frac{|\boldsymbol{\Sigma}_{x,v'}|}{|\boldsymbol{\Sigma}_{x,v}|} = \frac{|\boldsymbol{A}_i \boldsymbol{\Sigma}_{v'} \boldsymbol{A}_i^T|}{|\boldsymbol{A}_i \boldsymbol{\Sigma}_v \boldsymbol{A}_i^T|} = \frac{|\boldsymbol{R}_i^T \boldsymbol{Q}_i^T \boldsymbol{\Sigma}_{v'} \boldsymbol{Q}_i \boldsymbol{R}_i|}{|\boldsymbol{R}_i^T \boldsymbol{Q}_i^T \boldsymbol{\Sigma}_v \boldsymbol{Q}_i \boldsymbol{R}_i|}$$

$$= \frac{|\boldsymbol{R}_i|^2 |\boldsymbol{Q}_i^T \boldsymbol{\Sigma}_{v'} \boldsymbol{Q}_i|}{|\boldsymbol{R}_i|^2 |\boldsymbol{Q}_i^T \boldsymbol{\Sigma}_v \boldsymbol{Q}_i|} = \frac{|\boldsymbol{Q}_i^T \boldsymbol{\Sigma}_{v'} \boldsymbol{Q}_i|}{|\boldsymbol{Q}_i^T \boldsymbol{\Sigma}_v \boldsymbol{Q}_i|}$$

The $|\boldsymbol{R}_i|^2$ factors cancel, showing the determinant ratio depends only on $\boldsymbol{Q}_i^T$.

For the trace term, using the cyclic property of trace and $\boldsymbol{A}_i = \boldsymbol{R}_i^T \boldsymbol{Q}_i^T$:

$$\operatorname{tr}(\boldsymbol{\Sigma}_{x,v'}^{-1}\boldsymbol{\Sigma}_{x,v}) = \operatorname{tr}\left[(\boldsymbol{A}_i \boldsymbol{\Sigma}_{v'} \boldsymbol{A}_i^T)^{-1} \boldsymbol{A}_i \boldsymbol{\Sigma}_v \boldsymbol{A}_i^T\right]$$

$$= \operatorname{tr}\left[(\boldsymbol{R}_i^T \boldsymbol{Q}_i^T \boldsymbol{\Sigma}_{v'} \boldsymbol{Q}_i \boldsymbol{R}_i)^{-1} \boldsymbol{R}_i^T \boldsymbol{Q}_i^T \boldsymbol{\Sigma}_v \boldsymbol{Q}_i \boldsymbol{R}_i\right]$$

$$= \operatorname{tr}\left[\boldsymbol{R}_i^{-1} (\boldsymbol{Q}_i^T \boldsymbol{\Sigma}_{v'} \boldsymbol{Q}_i)^{-1} \boldsymbol{R}_i^{-T} \boldsymbol{R}_i^T \boldsymbol{Q}_i^T \boldsymbol{\Sigma}_v \boldsymbol{Q}_i \boldsymbol{R}_i\right]$$

$$= \operatorname{tr}\left[(\boldsymbol{Q}_i^T \boldsymbol{\Sigma}_{v'} \boldsymbol{Q}_i)^{-1} \boldsymbol{Q}_i^T \boldsymbol{\Sigma}_v \boldsymbol{Q}_i\right]$$

where we used cyclic permutation: $\operatorname{tr}(\boldsymbol{R}_i^{-1}\boldsymbol{B}\boldsymbol{R}_i) = \operatorname{tr}(\boldsymbol{B})$. Again, the $\boldsymbol{R}_i$ matrices cancel.

Therefore, the KL divergence depends only on the column space of $\boldsymbol{A}_i^T$ (i.e., $\boldsymbol{Q}_i$), not on the specific choice of basis within that space. We proceed assuming each $\boldsymbol{A}_i$ has orthonormal rows.

B.2.2. KL DIVERGENCE FOR COMPRESSED OBSERVATIONS

Define the projected hypotheses for iteration $i$:

$$\boldsymbol{w}_v = \boldsymbol{A}_i \bar{\boldsymbol{u}}_v, \quad \boldsymbol{w}_{v'} = \boldsymbol{A}_i \bar{\boldsymbol{u}}_{v'}$$

and the reduced covariance matrices:

$$\boldsymbol{\Sigma}_{x,v} = \lambda_2 \boldsymbol{I}_2 + \Delta\, \boldsymbol{w}_v \boldsymbol{w}_v^T, \qquad \boldsymbol{\Sigma}_{x,v'} = \lambda_2 \boldsymbol{I}_2 + \Delta\, \boldsymbol{w}_{v'} \boldsymbol{w}_{v'}^T$$

Define $\kappa = \Delta/\lambda_2$. Using the Woodbury identity:

$$\boldsymbol{\Sigma}_{x,v'}^{-1} = \frac{1}{\lambda_2}\left(\boldsymbol{I}_2 + \kappa\, \boldsymbol{w}_{v'} \boldsymbol{w}_{v'}^T\right)^{-1} = \frac{1}{\lambda_2}\left(\boldsymbol{I}_2 - \frac{\kappa}{1+\kappa\|\boldsymbol{w}_{v'}\|^2}\, \boldsymbol{w}_{v'} \boldsymbol{w}_{v'}^T\right)$$

The per-iteration KL divergence is:

$$D_{\mathrm{KL}}(P_{x,v}, P_{x,v'}) = \frac{1}{2}\left[\kappa\|\boldsymbol{w}_v\|^2 - \frac{\kappa\|\boldsymbol{w}_{v'}\|^2 + \kappa^2(\boldsymbol{w}_{v'}^T\boldsymbol{w}_v)^2}{1+\kappa\|\boldsymbol{w}_{v'}\|^2} + \ln\frac{1+\kappa\|\boldsymbol{w}_{v'}\|^2}{1+\kappa\|\boldsymbol{w}_v\|^2}\right]$$

### B.2.3. BOUNDING KL FOR ADJACENT HYPOTHESES

For adjacent hypotheses differing in coordinate $j$, we abbreviate $\boldsymbol{w} = \boldsymbol{w}_v$ and $\boldsymbol{w}' = \boldsymbol{w}_{v'}$ and decompose:

$$\boldsymbol{w} = \boldsymbol{A}_i \bar{\boldsymbol{u}}_{\text{base}} + \frac{\beta}{2} \boldsymbol{A}_i \boldsymbol{e}_{j+1}, \quad \boldsymbol{w}' = \boldsymbol{A}_i \bar{\boldsymbol{u}}_{\text{base}} - \frac{\beta}{2} \boldsymbol{A}_i \boldsymbol{e}_{j+1}$$

where $\bar{\boldsymbol{u}}_{\text{base}} = \boldsymbol{e}_1 \sqrt{1 - s\beta^2/4} + \frac{\beta}{2} \sum_{k \neq j} v[k] \boldsymbol{e}_{k+1}$ is the common component.

Define $\tilde{\boldsymbol{w}} = \boldsymbol{A}_i \bar{\boldsymbol{u}}_{\text{base}}$ and $\boldsymbol{z} = \boldsymbol{A}_i \boldsymbol{e}_{j+1}$. Let $m = \|\tilde{\boldsymbol{w}}\|^2$, $p = \tilde{\boldsymbol{w}}^T \boldsymbol{z}$, and $q = \|\boldsymbol{z}\|^2 = a_{i,j+1}^2 + b_{i,j+1}^2$ where $a_{i,j+1}, b_{i,j+1}$ are the $(j+1)$-th components of $\boldsymbol{a}_i, \boldsymbol{b}_i$.

We can compute:

$$\|\boldsymbol{w}\|^2 = m + \beta p + \frac{\beta^2 q}{4}, \quad \|\boldsymbol{w}'\|^2 = m - \beta p + \frac{\beta^2 q}{4}, \quad \boldsymbol{w}'^T \boldsymbol{w} = m - \frac{\beta^2 q}{4}$$

This gives $\|\boldsymbol{w}\|^2 - \|\boldsymbol{w}'\|^2 = 2\beta p$ and $\|\boldsymbol{w}\|^2 \|\boldsymbol{w}'\|^2 - (\boldsymbol{w}'^T \boldsymbol{w})^2 = \beta^2 (mq - p^2)$.

Substituting into the KL formula:

$$D_{KL} = \frac{1}{2} \left[ \frac{2\kappa\beta p + \kappa^2 \beta^2 (mq - p^2)}{1 + \kappa \|\boldsymbol{w}'\|^2} + \ln\left(1 - \frac{2\kappa\beta p}{1 + \kappa\|\boldsymbol{w}\|^2}\right) \right]$$

Using the inequality $\ln(1 - x) \leq -x$ for the logarithm term:

$$
\begin{aligned}
D_{KL} &\leq \frac{1}{2} \left[ \frac{2\kappa\beta p + \kappa^2 \beta^2 (mq - p^2)}{1 + \kappa\|\boldsymbol{w}'\|^2} - \frac{2\kappa\beta p}{1 + \kappa\|\boldsymbol{w}\|^2} \right] \\
&= \frac{1}{2} \left[ \frac{\kappa^2 \beta^2 (mq - p^2)}{1 + \kappa\|\boldsymbol{w}'\|^2} + 2\kappa\beta p \left( \frac{1}{1 + \kappa\|\boldsymbol{w}'\|^2} - \frac{1}{1 + \kappa\|\boldsymbol{w}\|^2} \right) \right] \\
&= \frac{1}{2} \left[ \frac{\kappa^2 \beta^2 (mq - p^2)}{1 + \kappa\|\boldsymbol{w}'\|^2} + 2\kappa\beta p \cdot \frac{\kappa(\|\boldsymbol{w}\|^2 - \|\boldsymbol{w}'\|^2)}{(1 + \kappa\|\boldsymbol{w}'\|^2)(1 + \kappa\|\boldsymbol{w}\|^2)} \right] \\
&= \frac{1}{2} \left[ \frac{\kappa^2 \beta^2 (mq - p^2)}{1 + \kappa\|\boldsymbol{w}'\|^2} + 2\kappa\beta p \cdot \frac{2\kappa\beta p}{(1 + \kappa\|\boldsymbol{w}'\|^2)(1 + \kappa\|\boldsymbol{w}\|^2)} \right] \\
&= \frac{1}{2} \left[ \frac{\kappa^2 \beta^2 (mq - p^2)}{1 + \kappa\|\boldsymbol{w}'\|^2} + \frac{4\kappa^2 \beta^2 p^2}{(1 + \kappa\|\boldsymbol{w}'\|^2)(1 + \kappa\|\boldsymbol{w}\|^2)} \right] \\
&\leq \frac{1}{2} \left[ \frac{\kappa^2 \beta^2 (mq - p^2)}{1 + \kappa\|\boldsymbol{w}'\|^2} + \frac{4\kappa^2 \beta^2 p^2}{1 + \kappa\|\boldsymbol{w}'\|^2} \right] = \frac{\kappa^2 \beta^2 (mq + 3p^2)}{2(1 + \kappa\|\boldsymbol{w}'\|^2)} \\
&\leq \frac{4\kappa^2 \beta^2 mq}{2(1 + \kappa\|\boldsymbol{w}'\|^2)} = \frac{2\kappa^2 \beta^2 mq}{1 + \kappa\|\boldsymbol{w}'\|^2}
\end{aligned}
$$

where the last inequality uses $p^2 \leq mq$ by Cauchy-Schwarz. To bound the ratio $\frac{m}{1+\kappa\|\boldsymbol{w}'\|^2}$, we use a monotonicity argument. Since $|p| \leq \sqrt{mq} \leq \sqrt{m}$ (as $q \leq 1$), we have:

$$\|\boldsymbol{w}'\|^2 = m - \beta p + \frac{\beta^2 q}{4} \geq m - \beta\sqrt{m}$$

Define $g(m) = \frac{m}{1+\kappa(m - \beta\sqrt{m})}$. Computing the derivative:

$$g'(m) = \frac{1 - \frac{\kappa\beta\sqrt{m}}{2}}{(1 + \kappa(m - \beta\sqrt{m}))^2}$$

The numerator is positive when $\kappa\beta\sqrt{m} < 2$. Since $m \leq 1$, this holds whenever $\kappa\beta < 2$. Substituting $\beta \leq \sqrt{2/(d-1)}$ and $\kappa = \Delta/\lambda_2$, the condition $\kappa\beta < 2$ is satisfied when $\Delta < \lambda_2\sqrt{2(d-1)}$.

Under this condition, $g(m)$ is monotonically increasing, so $g(m) \leq g(1) = \frac{1}{1+\kappa(1-\beta)}$. For $\beta \leq 1/2$ (which holds when $d \geq 9$), we have $g(1) \leq \frac{2}{1+\kappa}$. Therefore:

$$D_{\text{KL}}(P_{x,v}, P_{x,v'}) \leq \frac{2\kappa^2 \beta^2 mq}{1 + \kappa\|\boldsymbol{w}'\|^2} \leq \frac{4\kappa^2 \beta^2 q}{1 + \kappa} = \frac{4\Delta^2 \beta^2 q}{\lambda_1 \lambda_2} \leq \frac{8\Delta^2 \alpha^2}{\lambda_1 \lambda_2}(a_{i,j+1}^2 + b_{i,j+1}^2)$$

The KL divergence for distinguishing hypotheses differing in coordinate $j$ is proportional to $a_{i,j+1}^2 + b_{i,j+1}^2$ — the total energy of the measurement vectors in that coordinate at iteration $i$.

### B.2.4. AVERAGE-ENERGY ARGUMENT

Since $\boldsymbol{a}_i$ and $\boldsymbol{b}_i$ are unit vectors at each iteration $i$:

$$\sum_{j=1}^{d} a_{i,j}^2 = \|\boldsymbol{a}_i\|^2 = 1, \quad \sum_{j=1}^{d} b_{i,j}^2 = \|\boldsymbol{b}_i\|^2 = 1$$

so the per-iteration energy across the $s = d - 1$ perturbation coordinates is bounded pointwise:

$$\sum_{j=1}^{s} q_i^{(j)} \leq \sum_{j=1}^{d} (a_{i,j}^2 + b_{i,j}^2) = 2, \qquad q_i^{(j)} := a_{i,j+1}^2 + b_{i,j+1}^2$$

Because measurements may be adaptive, $\boldsymbol{a}_i, \boldsymbol{b}_i$ depend on $\boldsymbol{x}_1, \ldots, \boldsymbol{x}_{i-1}$ and the energies $q_i^{(j)}$ are random variables whose joint law depends on the underlying hypothesis. We therefore work with the expected per-coordinate energy under each hypothesis:

$$E_j(v) := \mathbb{E}_{P_v}\left[\sum_{i=1}^{t} q_i^{(j)}\right]$$

Since $\sum_j q_i^{(j)} \leq 2$ pointwise, $\sum_{j=1}^{s} E_j(v) \leq 2t$ for every $v \in V$.

### B.2.5. BOUNDING THE AVERAGED TOTAL VARIATION

By the chain rule for KL divergence, conditioning on the history $\boldsymbol{x}_1, \ldots, \boldsymbol{x}_{i-1}$ renders $\boldsymbol{a}_i, \boldsymbol{b}_i$ deterministic at iteration $i$, so the per-iteration bound $D_{\mathrm{KL},i} \leq (4\Delta^2\beta^2/\lambda_1\lambda_2)\, q_i^{(j)}$ derived above applies. Summing across iterations and taking expectations under $P_v$:

$$D_{\mathrm{KL}}\left(P_v^{(t)} \,\|\, P_{v^{(j)}}^{(t)}\right) \leq \frac{4\Delta^2\beta^2}{\lambda_1\lambda_2}\, E_j(v)$$

By Pinsker's inequality:

$$\mathrm{TV}\left(P_v^{(t)}, P_{v^{(j)}}^{(t)}\right) \leq \sqrt{\frac{1}{2} D_{\mathrm{KL}}\left(P_v^{(t)} \,\|\, P_{v^{(j)}}^{(t)}\right)} \leq \sqrt{\frac{2\Delta^2\beta^2}{\lambda_1\lambda_2}\, E_j(v)}$$

To bound the averaged total variation appearing in Lemma B.1, we apply Jensen's inequality (concavity of $\sqrt{\cdot}$) twice — first over hypotheses with $v[j] = 1$, then over coordinates $j$:

$$\overline{\mathrm{TV}} := \frac{1}{s}\sum_{j=1}^{s}\frac{1}{2^{s-1}}\sum_{\substack{v \in V \\ v[j]=1}} \mathrm{TV}\left(P_v^{(t)}, P_{v^{(j)}}^{(t)}\right) \leq \sqrt{\frac{2\Delta^2\beta^2}{s\lambda_1\lambda_2}\sum_{j=1}^{s}\bar{E}_j}$$

where $\bar{E}_j := \frac{1}{2^{s-1}}\sum_{v:\,v[j]=1} E_j(v)$. Enlarging the sum over $\{v : v[j] = 1\}$ to all of $V$ and using $\sum_j E_j(v) \leq 2t$ pointwise in $v$:

$$\sum_{j=1}^{s}\bar{E}_j \leq \frac{1}{2^{s-1}}\sum_{v \in V}\sum_{j=1}^{s} E_j(v) \leq \frac{2^s \cdot 2t}{2^{s-1}} = 4t$$

Therefore:

$$\overline{\mathrm{TV}} \leq \sqrt{\frac{8\Delta^2\beta^2 t}{s\lambda_1\lambda_2}} = \gamma\beta, \qquad \gamma := \sqrt{\frac{8\Delta^2 t}{s\lambda_1\lambda_2}}$$

B.2.6. FINAL LOWER BOUND

Substituting into Lemma B.1 with the Hamming separation $\min_{H \geq 1} \rho^2/H \geq \beta^2/2$ established in the Hypercube Construction:

$$\inf_{\hat{\boldsymbol{u}}_t} \max_{v \in V} \mathbb{E}_{P_v} \rho(\hat{\boldsymbol{u}}_t, \bar{\boldsymbol{u}}_v)^2 \;\geq\; \frac{s}{8} \cdot \frac{\beta^2}{2} \cdot (1 - \gamma\beta) \;=\; \frac{s\beta^2}{16}(1 - \gamma\beta)$$

Maximizing $\beta^2(1 - \gamma\beta)$ over $\beta > 0$ yields $\beta^* = 2/(3\gamma)$ with $\gamma\beta^* = 2/3$ and $(1 - \gamma\beta^*) = 1/3$. Hence:

$$(\beta^*)^2 \;=\; \frac{4}{9\gamma^2} \;=\; \frac{(d-1)\lambda_1\lambda_2}{18\Delta^2 t}$$

For the construction to be valid we require $(\beta^*)^2 \leq 2/s$. With $s = d - 1$, this imposes:

$$t \;\geq\; \frac{(d-1)^2\lambda_1\lambda_2}{36\Delta^2}$$

This sample complexity is precisely the regime where adaptive compressed sensing requires $\Theta(d)$ more samples than full observation. Substituting $\beta^*$ into the lower bound:

$$\text{Lower Bound} \;=\; \frac{s(\beta^*)^2}{16} \cdot \frac{1}{3} \;=\; \frac{s}{48} \cdot \frac{4}{9\gamma^2} \;=\; \frac{s}{108\gamma^2} \;=\; \frac{(d-1)^2\lambda_1\lambda_2}{864\Delta^2 t}$$

Therefore:

$$\boxed{\inf_{\hat{\boldsymbol{u}}_t} \max_{v \in V} \mathbb{E}_{P_v} \left[1 - (\hat{\boldsymbol{u}}_t^T \bar{\boldsymbol{u}}_v)^2\right] \;\geq\; \frac{(d-1)^2\lambda_1\lambda_2}{864\Delta^2 t}} \tag{18}$$

This establishes the minimax lower bound for adaptive compressed sensing with 2 measurements per iteration. The constant $1/864$ comes from optimizing over the perturbation scale $\beta$ and reflects the inherent looseness from the proof technique.

The bound is valid for $t \geq \frac{(d-1)^2\lambda_1\lambda_2}{36\Delta^2}$, $d \geq 9$, and $\Delta < \lambda_2\sqrt{2(d-1)}$. These conditions are mild: $d \geq 9$ is satisfied for any non-trivial problem, and $\Delta < \lambda_2\sqrt{2(d-1)}$ covers all regimes where $\Delta = O(\lambda_2\sqrt{d})$. This establishes that the $d^2$ dimension dependence is information-theoretically necessary when restricted to two adaptive measurements per iteration.

## B.3. Non-Adaptive Compressed Measurements Lower Bound

We now extend the analysis to the *non-adaptive* setting, in which the measurement matrices $\boldsymbol{A}_1, \ldots, \boldsymbol{A}_t \in \mathbb{R}^{2 \times d}$ are fixed before any data is observed: $\boldsymbol{A}_i$ does not depend on $\boldsymbol{v}_1, \ldots, \boldsymbol{v}_{i-1}$ or on the underlying hypothesis. As in the adaptive case, we may assume without loss of generality that each $\boldsymbol{A}_i$ has orthonormal rows. Since $\{\boldsymbol{A}_i\}$ is fixed before data, the lower-bound prover can rotate the hypothesis family to align $\bar{\boldsymbol{u}}$ with the least-measured direction, gaining an additional factor of $d$ in sample complexity.

B.3.1. SIGNAL CAPTURE AND ADVERSARIAL ROTATION

The key new ingredient is a lemma showing that any non-adaptive measurement scheme leaves some direction poorly measured.

**Lemma B.2** (Signal Capture). *For any non-adaptive measurement scheme with $\boldsymbol{A}_1, \ldots, \boldsymbol{A}_t \in \mathbb{R}^{2 \times d}$ having orthonormal rows, there exists a unit vector $\bar{\boldsymbol{u}} \in \mathcal{S}^{d-1}$ such that:*

$$\sum_{i=1}^{t} \|\boldsymbol{A}_i \bar{\boldsymbol{u}}\|^2 \;\leq\; \frac{2t}{d}$$

*Proof.* The measurement energy matrix $\boldsymbol{M} := \sum_{i=1}^{t} \boldsymbol{A}_i^T \boldsymbol{A}_i$ satisfies $\operatorname{tr}(\boldsymbol{M}) = \sum_i \operatorname{tr}(\boldsymbol{A}_i \boldsymbol{A}_i^T) = 2t$, since each $\boldsymbol{A}_i$ has orthonormal rows ($\boldsymbol{A}_i \boldsymbol{A}_i^T = \boldsymbol{I}_2$). Hence $\mu_{\min}(\boldsymbol{M}) \leq \operatorname{tr}(\boldsymbol{M})/d = 2t/d$. Choosing $\bar{\boldsymbol{u}}$ as the corresponding eigenvector gives $\sum_i \|\boldsymbol{A}_i \bar{\boldsymbol{u}}\|^2 = \bar{\boldsymbol{u}}^T \boldsymbol{M} \bar{\boldsymbol{u}} = \mu_{\min}(\boldsymbol{M}) \leq 2t/d$. $\square$

The adversary exploits this lemma as follows. Write $M = Q \operatorname{diag}(\mu_1, \ldots, \mu_d) Q^T$ with $\mu_1 \leq \cdots \leq \mu_d$. By the rotational invariance of the construction (see the problem setup in Appendix B), we may rotate coordinates so that $e_1$ aligns with the eigenvector of $\mu_{\min}(M)$. In this frame, the hypercube construction places its base direction $e_1$ along the least-measured direction, yielding $\mu_1 \leq 2t/d$ in the rotated basis. This rotation is valid *only because* $\{A_i\}_{i=1}^t$ is fixed — the adversary chooses the rotation after seeing the measurements.

### B.3.2. RADEMACHER AVERAGING OF SIGNAL CAPTURE

For adjacent hypotheses differing in coordinate $j$, the per-iteration KL bound from the previous derivation (raw form, before applying the monotonicity bound $g(m) \leq g(1)$) reads:

$$D_{\mathrm{KL},i} \leq \frac{2\kappa^2 \beta^2 m_i(v) q_i^{(j)}}{1 + \kappa \|w'\|^2} \leq 2\kappa^2 \beta^2 m_i(v) q_i^{(j)}, \qquad \kappa = \Delta/\lambda_2$$

where $m_i(v) := \|A_i \bar{u}_{\mathrm{base}}(v)\|^2$ is the signal capture under hypothesis $v$ (with $\bar{u}_{\mathrm{base}}(v) = e_1 \sqrt{1 - s\beta^2/4} + \frac{\beta}{2} \sum_{k \neq j} v[k] e_{k+1}$ the common component across the adjacent pair), and $q_i^{(j)} = a_{i,j+1}^2 + b_{i,j+1}^2$ is the perturbation energy. The non-adaptive case uses this looser (denominator-dropped) form: it retains the signal capture factor $m_i(v)$, which combined with the signal-capture lemma above enables the extra factor of $d$. The factor cannot be exploited in the adaptive case, where $A_i$ depends on the data — hence $m_i(v)$ becomes statistically coupled to the hypothesis through the history, and the Rademacher cancellation below fails.

**Lemma B.3** (Rademacher Averaging). *Under the adversarial rotation of Lemma B.2, for any non-adaptive measurement scheme with $\mu_1 := \sum_i [B_i]_{11} \leq 2t/d$ (where $B_i := A_i^T A_i$) and $\beta^2 \leq 2/(d-1)$:*

$$\sum_{j=1}^{s} \sum_{i=1}^{t} \bar{m}_i^{(j)} q_i^{(j)} \leq \frac{8t}{d}$$

*where $\bar{m}_i^{(j)} := \frac{1}{2^{s-1}} \sum_{v : v[j]=1} m_i(v)$.*

*Proof.* Expanding the quadratic form $m_i(v) = \bar{u}_{\mathrm{base}}(v)^T B_i \bar{u}_{\mathrm{base}}(v)$:

$$m_i(v) = \left(1 - \tfrac{s\beta^2}{4}\right) [B_i]_{11} + \beta \sqrt{1 - \tfrac{s\beta^2}{4}} \sum_{k \neq j} v[k] [B_i]_{1,k+1} + \tfrac{\beta^2}{4} \sum_{k,\ell \neq j} v[k]v[\ell] [B_i]_{k+1,\ell+1}$$

Under the uniform average over $\{v : v[j] = 1\}$, the free coordinates $v[k]$ ($k \neq j$) are independent Rademacher variables. The cross terms vanish since $\mathbb{E}[v[k]] = 0$, and the perturbation–perturbation terms reduce to the diagonal since $\mathbb{E}[v[k]v[\ell]] = \delta_{k\ell}$:

$$\bar{m}_i^{(j)} = \left(1 - \tfrac{s\beta^2}{4}\right) [B_i]_{11} + \tfrac{\beta^2}{4} \sum_{k \neq j} [B_i]_{k+1,k+1} \leq [B_i]_{11} + \tfrac{\beta^2}{2}$$

where the inequality uses $\sum_{k \neq j} [B_i]_{k+1,k+1} \leq \operatorname{tr}(B_i) = 2$. Since this bound is independent of $j$, writing $R_i := \sum_{j=1}^{s} q_i^{(j)} \leq \operatorname{tr}(B_i) = 2$:

$$\sum_{j=1}^{s} \sum_{i=1}^{t} \bar{m}_i^{(j)} q_i^{(j)} \leq \sum_{i=1}^{t} \left([B_i]_{11} + \tfrac{\beta^2}{2}\right) R_i \leq 2\left(\mu_1 + \tfrac{\beta^2 t}{2}\right) \leq 2\left(\tfrac{2t}{d} + \tfrac{t}{d-1}\right) \leq \tfrac{8t}{d}$$

where $\mu_1 \leq 2t/d$ by Lemma B.2 (in the rotated basis) and $\beta^2 \leq 2/(d-1)$. $\qquad\square$

### B.3.3. FINAL NON-ADAPTIVE LOWER BOUND

Because the measurement matrices are deterministic, the per-iteration KL bound sums directly across iterations: $D_{\mathrm{KL}}(P_v^{(t)} \| P_{v^{(j)}}^{(t)}) \leq 2\kappa^2 \beta^2 \sum_i m_i(v) q_i^{(j)}$. By Pinsker's inequality:

$$\mathrm{TV}\left(P_v^{(t)}, P_{v^{(j)}}^{(t)}\right) \leq \sqrt{\tfrac{1}{2} D_{\mathrm{KL}}} \leq \kappa\beta \sqrt{\sum_{i=1}^{t} m_i(v) q_i^{(j)}}$$

Averaging via Jensen's inequality (concavity of $\sqrt{\cdot}$), first over hypotheses with $v[j] = 1$ and then over coordinates $j$:

$$\overline{\mathrm{TV}}^2 \;\leq\; \frac{\kappa^2\beta^2}{s} \sum_{j=1}^{s} \sum_{i=1}^{t} \bar{m}_i^{(j)} \, q_i^{(j)} \;\leq\; \frac{8\kappa^2\beta^2 t}{sd}$$

where the last inequality is Lemma B.3. Hence $\overline{\mathrm{TV}} \leq \gamma_A \beta$ with $\gamma_A := \kappa\sqrt{8t/(sd)}$.

Substituting into Lemma B.1 with the Hamming separation $\min_{H \geq 1} \rho^2/H \geq \beta^2/2$:

$$\inf_{\hat{\boldsymbol{u}}_t} \max_{v \in V} \mathbb{E}_{P_v} \rho(\hat{\boldsymbol{u}}_t, \bar{\boldsymbol{u}}_v)^2 \;\geq\; \frac{s\beta^2}{16}(1 - \gamma_A \beta)$$

Maximizing $\beta^2(1 - \gamma_A \beta)$ yields $\beta^* = 2/(3\gamma_A)$, $(1 - \gamma_A\beta^*) = 1/3$, and:

$$(\beta^*)^2 \;=\; \frac{4}{9\gamma_A^2} \;=\; \frac{(d-1)\,d\,\lambda_2^2}{18\,\Delta^2\,t}$$

Validity $((\beta^*)^2 \leq 2/s)$ requires $t \geq (d-1)^2 d\,\lambda_2^2/(36\Delta^2)$. Substituting:

$$\text{Lower Bound} \;=\; \frac{s(\beta^*)^2}{48} \;=\; \frac{s}{108\,\gamma_A^2} \;=\; \frac{(d-1)^2 d\,\lambda_2^2}{864\,\Delta^2\,t}$$

Therefore:

$$\boxed{\inf_{\hat{\boldsymbol{u}}_t} \max_{v \in V} \mathbb{E}_{P_v}\left[1 - (\hat{\boldsymbol{u}}_t^T \bar{\boldsymbol{u}}_v)^2\right] \;\geq\; \frac{(d-1)^2 d\,\lambda_2^2}{864\,\Delta^2\,t}} \tag{19}$$

This establishes the minimax lower bound for non-adaptive compressed sensing with two measurements per iteration. The headline contribution is the dimension scaling: *non-adaptive measurements require an additional factor of $d$ in sample complexity beyond the adaptive bound*, reflecting the loss of signal capture under the adversarial rotation. This $d^3$ scaling holds in *all* regimes of $(\lambda_1, \lambda_2)$ — the dimension-dependence gap between adaptive and non-adaptive is unconditional. The bound is valid for $t \geq (d-1)^2 d\,\lambda_2^2/(36\Delta^2)$; unlike the adaptive case, no separate constraint on $\Delta/\lambda_2$ is required, since the non-adaptive proof operates directly on the raw KL bound without invoking the $g(m) \leq g(1)$ monotonicity step that yields the $\kappa\beta < 2$ condition.

The eigenvalue dependence $\lambda_2^2$ (rather than $\lambda_1\lambda_2$ as in the adaptive case) reflects that in the poor-signal-capture regime ($m_i \approx 1/d$) the projected $2 \times 2$ covariance has both eigenvalues $\approx \lambda_2$ and $\lambda_1$ does not enter the projected geometry. In the hard estimation regime $\Delta \leq \lambda_2$ (equivalently $\lambda_1 \leq 2\lambda_2$), we have $\lambda_2^2 \geq \lambda_1\lambda_2/2$, so the non-adaptive eigenvalue scaling matches the adaptive $\lambda_1\lambda_2$ up to a factor of 2.

Heuristically, the $\lambda_2^2$ dependence is minimax-tight in the worst-case orientation. Under the adversarial rotation the projected $2 \times 2$ covariance has $\det \boldsymbol{\Sigma}_x \approx \lambda_2^2$ (versus $\lambda_1\lambda_2$ when the signal is captured), and the projected Fisher information accordingly scales as $1/\lambda_2^2$. The same projection accounts for the factor of $d$ between the adaptive and non-adaptive bounds. The per-sample Cramér–Rao variance is $d^2$ times worse in this worst-case orientation, and Lemma B.2's energy budget restricts the informative-measurement fraction to $\sim 1/d$. The net penalty is $d$.

In the hard regime $\Delta \leq \lambda_2$, the bound takes the $\lambda_1\lambda_2$ form, matching the eigenvalue scaling of the adaptive bound (Theorem 4.2):

**Corollary B.4** (Non-Adaptive Lower Bound, Hard Regime)**.** *Under the conditions of the non-adaptive lower bound established above and additionally $\Delta \leq \lambda_2$ (equivalently $\lambda_1 \leq 2\lambda_2$):*

$$\boxed{\inf_{\hat{\boldsymbol{u}}_t} \max_{v \in V} \mathbb{E}_{P_v}\left[1 - (\hat{\boldsymbol{u}}_t^T \bar{\boldsymbol{u}}_v)^2\right] \;\geq\; \frac{(d-1)^2 d\,\lambda_1\lambda_2}{1728\,\Delta^2\,t}} \tag{20}$$

*Proof.* The hypothesis $\Delta \leq \lambda_2$ implies $\lambda_1 = \Delta + \lambda_2 \leq 2\lambda_2$, hence $\lambda_2^2 \geq \lambda_1\lambda_2/2$. Substituting into (19) yields the claim. $\square$

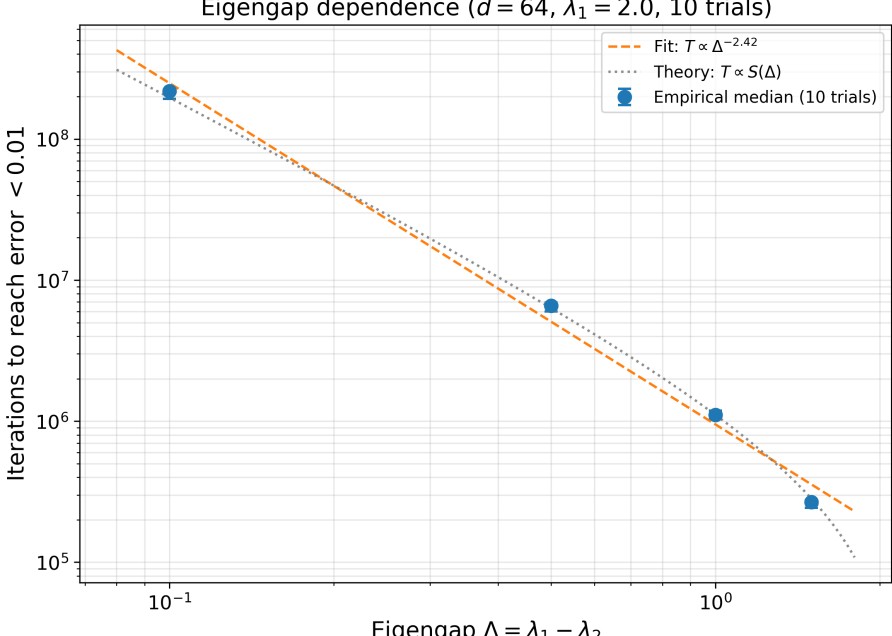

*Figure 4.* Convergence dependence on the eigengap $\Delta = \lambda_1 - \lambda_2$. Fixing $d = 64$ and $\lambda_1 = 2$, we vary $\lambda_2 \in \{0.5, 1.0, 1.5, 1.9\}$ so that $\Delta \in \{1.5, 1.0, 0.5, 0.1\}$. Markers show median iterations to reach error $10^{-2}$ over 10 trials. Reference curves: empirical least-squares fit $T \propto \Delta^{-2.42}$ (orange dashed) and theoretical $T \propto S(\Delta) = 3\lambda_1\lambda_2 d^2/\Delta^2 + 15\lambda_1 d/\Delta$ (gray dotted, anchored at $\Delta = 1$). The empirical points hug the $T \propto S(\Delta)$ curve, demonstrating that the parameter dependence in Theorem 4.1 captures the data accurately.

## C. Additional Experiments

### C.1. Eigengap Variation

To probe the eigengap dependence predicted by Theorem 4.1, we hold $d = 64$ and $\lambda_1 = 2$ fixed and vary $\lambda_2 \in \{0.5, 1.0, 1.5, 1.9\}$, giving $\Delta \in \{1.5, 1.0, 0.5, 0.1\}$. Figure 4 reports iterations to target error $10^{-2}$ on log-log axes. All 10 trials at every setting converged — including the most demanding case $\Delta = 0.1$, where the eigengap is more than an order of magnitude below the noise floor $\lambda_2 = 1.9$, and the median run takes $2.19 \times 10^8$ iterations.

The empirical log-log fit slope is $-2.42$. To understand this value, observe that the empirical points are nearly proportional to the theoretical complexity $S(\Delta) = 3\lambda_1\lambda_2 d^2/\Delta^2 + 15\lambda_1 d/\Delta$: the proportionality constant $T/S$ is approximately $40, 42, 44, 47$ across the four $\Delta$ values (the upper bound is loose by a roughly uniform factor of $\approx 40$–$50$). Because $\lambda_2 = \lambda_1 - \Delta$ varies with $\Delta$, the log-log slope of $S(\Delta)$ itself is not $-2$ but rather $-2.36$ on the $\{1.5, 1.0, 0.5, 0.1\}$ grid (computed analytically); the residual $-0.06$ to the empirical $-2.42$ comes from the small drift in the $T/S$ ratio. As $\Delta \to 0$ the $3\lambda_1\lambda_2 d^2/\Delta^2$ term dominates and $\lambda_2 \to \lambda_1$, so the local slope of $S(\Delta)$ approaches $-2$ (analytically: $-4.24, -2.85, -2.30, -2.05$ at our four $\Delta$ values). Thus the $1/\Delta^2$ eigengap dependence predicted by Theorems 4.1 and 4.2 is confirmed across more than an order of magnitude in $\Delta$, with the deviation from the naive $-2$ slope explained by the lower-order $\lambda_1 d/\Delta$ term in $S$ and the protocol's $\lambda_2(\Delta)$ dependence.

