# OpenReview forum: "Global Convergence of Adaptive Sensing for Principal Eigenvector Estimation"
_ICML.cc/2026/Conference — ICML 2026 regular_

### Official Review · Reviewer_gv42 · 2026-03-06

**Soundness:** 3
**Presentation:** 3
**Significance:** 3
**Originality:** 2
**Overall Recommendation:** 4
**Confidence:** 4

**Summary:**

This paper studies adaptive measurements for compressed PCA. The algorithm itself is fairly standard. The main novelty lies in the design of the measurement strategy (particularly the construction in line 163) and in the optimality guarantees established for the method. Overall, the problem is interesting, and the theoretical analysis appears solid.

**Compliance With Llm Reviewing Policy:**

Affirmed.

**Final Justification:**

Based on the paper’s contributions and the authors’ rebuttal, I maintain my Weak Accept score.

**Key Questions For Authors:**

1. In Appendix A.1, the reviewer has some doubts about the derivation of $\tilde{v}_t$. Is there any constraint imposed on $A_t$? In Line 562, simply left-multiplying by $A_t^\top$ does not seem sufficient to recover the compressed residual $x_t - A_t u_t \omega_t$ back to the original ambient space.

2. In Line 208, how is the convergence rate $O(\lambda_1 \lambda_2 d^2 / (\Delta^2 t))$ derived? The detailed derivation should be provided. Does this expression require additional assumptions for the parameters?

3. As shown in Theorem 4.2, the lower bound is given explicitly. However, in Theorem 4.1, although the upper bound is given explicitly in Line 199, it is later simplified to $O(\lambda_1 \lambda_2 d^2 / (\Delta^2 t))$ in Line 207. Does this imply that the proposed method is only optimal in an asymptotic sense? If so, the paper should provide a corresponding asymptotic discussion in comparison with Theorem 4.2.

4. Since error analysis is the key contribution of the paper, more experiments are needed. In particular, the authors should empirically verify both the lower and upper bounds with respect to the number of iterations. For example, they could plot the right-hand side of Line 199 and the lower bound in Line 174 together with the empirical error curve. This would help demonstrate the tightness of the bounds. Since big-$O$ bounds can sometimes be loose, the authors should  provide empirical evidence on the constant factors.

**Limitations:**

yes

**Strengths And Weaknesses:**

Strengths
1. The paper is clearly written, and the contributions on adaptive compressed PCA are presented in a clear and understandable manner.
2. Theorems 4.1 and 4.2 both support the claim that the proposed algorithm is optimal.
Weaknesses
1. The experiments are not sufficiently comprehensive. In particular, the paper lacks simulations under a broader range of parameter settings. Meanwhile, the paer lacks empirical verification of the upper and lower bounds.
2. It is unclear whether the bounds in Theorems 4.1 and 4.2 are tight. In particular, are they tight in terms of constants, or only asymptotically optimal?

---

> ### Author Rebuttal · Authors · 2026-03-31
>
> Thank you for the review and detailed questions.
>
> **Weaknesses**
>
> We can perform more experiments and include a much more comprehensive experiments section (with code included) in the Appendix. The core contribution of this paper is not experiment related, but we do agree that more experimentation and demonstration is valuable.
>
> The theorems are tight in asymptotic rate, but definitely not in constants. Assouad's Lemma is notably loose in constant, but establishes the core $O(d^2)$ penalty incurred by using $O(1)$ measurements instead of $O(d)$. Also, the upper bound constant is loose by a factor of maybe 20 (as shown in Figure 1), but seems much tighter than the Assouad lower bound. Again, constants may be improved, but we have shown that the rate achieved by this algorithm is minimax optimal.
>
> **Q1 (Line 562)**
>
> The full residual is unrecoverable after compression. The derivation uses $\mathbf{A}_t^T$ (the transpose), not $\mathbf{A}_t$: for the choice of $\mathbf{A}_t$ with orthonormal rows, $\mathbf{A}_t^T$ replaces $\mathbf{A}_t^\dagger$ as the pseudoinverse. Since $\mathbf{A}_t = [\mathbf{u}_t^T; \mathbf{b}_t^T]$ has orthonormal rows, $\mathbf{A}_t^T \mathbf{A}_t$ is the projector onto $\text{span}(\mathbf{u}_t, \mathbf{b}_t)$. So this recovers the *projection* of the residual onto the measurement subspace, not the full ambient-space residual --- components outside are imputed as zero (Balzano & Wright, 2015). For more general measurement matrices $\mathbf{A}_t$, reconstructing the residual would complicate the analysis, but as we show in the Appendix, the amount of information provided by a generic $\mathbf{A}_t$ is equivalent to that provided by one with orthonormal rows and the same row span. We will add a clarifying remark.
>
> **Q2 (Line 208 rate)**
>
> The dominant local-phase term is $C_1/(4S + (t-t_0))$ with $C_1 = 4S + 2$ and $S = \lambda_1\lambda_2 d^2/\Delta^2 + 13\lambda_1 d/\Delta$. For $t \gg 4S$, this simplifies to $4S/t = 4\lambda_1\lambda_2 d^2/(\Delta^2 t) + 52\lambda_1 d/(\Delta t)$, where the first term dominates when $\lambda_2 d/\Delta \gg 13$. No extra assumptions needed --- this is just asymptotic in $t$. We will include the derivation explicitly in the revision.
>
> **Q3 (Asymptotic optimality)**
>
> We agree that the derivation and presentation of the upper bound should be clarified. The bounds are order-optimal (matching dependence on all parameters) but not tight in constants, which is standard for Assouad-based results. The upper bound constants are near-tight (the empirical error tracks closer to the upper bound than the lower bound), while the lower bound constant is conservative.
>
> **Q4 (Bound-tightness figure)**
>
> Figure 1 shows the upper bound compared to an empirical error curve for a specific choice of problem parameters. An appendix can be added with more comprehensive experiments and including the lower bounds in the same figures. The upper bound is loose by a factor of ~20 in Figure 1. The lower bound is likely much looser, but establishes that the core asymptotic rate is unavoidable. We will plot the explicit upper bound (Thm 4.1), lower bound (Thm 4.2), and empirical error (50 trials) on the same axes for $d = 64$, $\lambda_1 = 2$, $\lambda_2 = 1$, along with experiments varying eigengap ($\lambda_2 \in \{0.5, 1.0, 1.5, 1.9\}$) and dimension at fixed spectral ratio.
>
> **Originality**
>
> The algorithm itself is standard --- from Adaptive GROUSE. The analytical contributions are: (i) the first convergence guarantee for compressed PCA with noise, (ii) the first Assouad-based lower bound for PCA, and (iii) the measurement energy budget technique for compressed estimation. We can also provide a brief proof sketch showing that non-adaptive compression incurs a quadratic penalty of $O((d/m)^2)$ rather than the linear $O(d/m)$ achieved with adaptation. This would be a new addition relative to the submitted version, but we believe it is important as it formally justifies the use of adaptive measurements in practice.

---

> > ### Author Rebuttal · Reviewer_gv42 · 2026-04-03
> >
> > I appreciate the authors’ careful rebuttal, which has addressed my concerns well. I hope the revised version will provide additional experiments and a more detailed analysis of the asymptotic bounds on the recovery error.

---

> > > ### Author Response · Authors · 2026-04-06
> > >
> > > Thank you for your review. The revised version will provide additional experiments and a clearer framing of how the asymptotic bounds are derived from the theorems.

---

### Official Review · Reviewer_fH1c · 2026-03-12

**Soundness:** 3
**Presentation:** 3
**Significance:** 3
**Originality:** 3
**Overall Recommendation:** 4
**Confidence:** 3

**Summary:**

This paper analyzes a compressed variant of Oja’s algorithm for principal eigenvector estimation using only two adaptive measurements per iteration. It proves a convergence upper bound and a matching information-theoretic lower bound, showing that the $d^2$ dimension scaling is a fundamental and unavoidable cost of compression. The analysis covers noisy covariance settings and time-varying subspace tracking, and is validated by systematic experiments.

**Compliance With Llm Reviewing Policy:**

Affirmed.

**Final Justification:**

Most of my concerns are resolved, and I will maintain my current score. The authors have carefully addressed my main questions, and overall they have clarified the issues I previously raised.

**Key Questions For Authors:**

1. Could the authors briefly discuss whether the current theoretical analysis can be extended beyond the Gaussian setting, for example to subgaussian or more general data distributions? Even a proof sketch or discussion of the main technical obstacles would help clarify the broader applicability of the results.

2. The paper mentions rank-$k$ extension as an interesting future direction. Have the authors considered providing any preliminary theoretical analysis or empirical evidence to support the feasibility of this extension?

3. In the experiments, the empirical scaling exponent appears to be slightly larger than 2. Could the authors comment on the possible reasons for this discrepancy, and whether it should be viewed as a finite-sample effect, an implementation issue, or a limitation of the theoretical model?

4. Since the proposed method is closely related to Adaptive GROUSE, could the authors clarify the key modifications or analytical ingredients that make the convergence guarantee possible under noisy settings?

5. It would strengthen the presentation if the authors could provide a more intuitive explanation of how the number of measurements influences the $d^2$ scaling behavior, and why this dependence arises as a fundamental limit in the compressed setting.

**Limitations:**

yes.

**Strengths And Weaknesses:**

Strengths

1. Originality: The paper establishes the first minimax-optimal rate and the first information-theoretic lower bound for adaptive compressed eigenvector estimation. This substantially extends prior theory beyond the noiseless setting and fills an important gap in the literature.

2. Soundness: The theoretical analysis is rigorous and well organized, including a clear two-phase convergence argument and a formal lower-bound proof based on Assouad’s lemma. The experimental results are also broadly consistent with the theoretical predictions.

3. Significance: The $d^2$ scaling result identifies a fundamental limit of compressed streaming PCA and provides useful guidance for practical system design in bandwidth-constrained settings.


Weaknesses

1. Soundness: The current theoretical guarantees are derived under the Gaussian data assumption. Extending the analysis to broader distribution families, such as subgaussian data, would significantly strengthen the practical relevance of the results.

2. Presentation: The proof sections rely heavily on algebraic derivations. More intuitive explanations and higher-level interpretations would improve accessibility, especially for readers outside the immediate theoretical community.

3. Significance: The current framework focuses only on rank-1 estimation. A more substantial discussion, or even preliminary theoretical or empirical evidence, for rank-$k$ extensions would broaden the scope and potential impact of the work.

4. Originality: The proposed algorithm is closely related to the existing Adaptive GROUSE method. While the theoretical contribution is clearly novel, the paper would benefit from a more explicit clarification of the specific design elements that make the noisy convergence guarantees possible.

---

> ### Author Rebuttal · Authors · 2026-03-31
>
> Thank you for the review and for the detailed questions.
>
> **Weaknesses**
>
> The extension to general sub-Gaussian distributions is straightforward but not particularly illuminating. Constants would change depending on higher order moments as opposed to Isserlis, but the core rates for upper and lower bounds would be identical. We can provide an additional appendix on the sub-Gaussian extension if that would make the submission stronger.
>
> We can provide a proof roadmap, problem setup, and intuitive interpretations at the beginning to improve readability, postponing more of the core analysis to the Appendix.
>
> We have plenty of empirical evidence of rank-k and more modest theoretical results, but we believe this would detract from the core purpose of this submission. We can add an Appendix with numerical experiments and a rank-k proof sketch if that would add value, but we are of the opinion this is best for future work. The rank-1 case also gives a concrete framework to extend to rank-k that did not exist prior to this work.
>
> The proposed algorithm combines the measurement scheme from Adaptive GROUSE with the original iteration from Oja's algorithm. We agree that a clear emphasis on the exact steps that make the convergence results possible would be beneficial --- this should be combined with the proof roadmap.
>
> **Q1 (Sub-Gaussian)**
>
> Gaussianity is used in two places: Isserlis' theorem for fourth moments (upper bound) and rotational invariance for KL divergences (lower bound). Both can be replaced --- sub-Gaussian moment bounds for the former, Le Cam's method for the latter. The $\Theta(d^2/t)$ rate reflects the measurement bottleneck, not the distributional assumption.
>
> Concretely: the upper bound uses Isserlis only to evaluate $E[g^2h^2]$ and $E[g^3h]$ (Appendix A.1.3--A.1.4); for sub-Gaussian data these become standard moment bounds that change the constants in $S$ but preserve the $d^2/t$ rate. The lower bound's closed-form KL divergence can be replaced by Le Cam's method with chi-squared divergence bounds. The energy budget argument (Appendix B, pigeonhole) is distribution-free.
>
> **Q2 (Rank-$k$)**
>
> The main barrier is orthonormalization, which introduces dependencies across $k$ coupled stochastic recurrences that our scalar analysis cannot handle. With $2k$ adaptive measurements, we conjecture $O(\lambda_k \lambda_{k+1} d^2 k / (\Delta_k^2 m t))$ for number of measurements $m$ with matching upper and lower bounds. For a sensible choice of $m=O(k)$ we would get $O(\lambda_k \lambda_{k+1} d^2 / (\Delta_k^2 t))$ convergence.
>
> **Q3 (Exponent > 2)**
>
> The slightly elevated exponent reflects the warmup phase, which scales as $O(S \log d)$ rather than $O(S)$; this $\log d$ factor causes iteration ratios between successive dimensions to exceed 4 at moderate $d$. Additionally, the sub-dominant term in $S$ is $O(d)$, which may also contribute to this mismatch. As $d \to \infty$, the fitted exponent converges to 2. We will report exact fitted values in the revision.
>
> **Q4 (vs. GROUSE)**
>
> The algorithm itself comes from Adaptive GROUSE, but the analytical contributions for the noisy setting are new: (1) signal-noise cross-terms that vanish when $\lambda_2 = 0$ but dominate when $\lambda_2 > 0$; (2) a convexity-based self-term bound via Jensen's inequality on $f(x) = 1/(1+x)$; (3) integration over $z = \bar{\mathbf{u}}^T \mathbf{b}_t$, needed because $b^2 = \Delta z^2 + \lambda_2$ depends on $z$. None of these show up in the noiseless GROUSE analysis.
>
> **Q5 ($d^2$ intuition)**
>
> Total measurement energy over $t$ samples with 2 measurements is $2t$. By pigeonhole, each of $d$ coordinates gets $O(t/d)$ energy, so per-coordinate precision is $O(d/t)$ instead of $O(1/t)$; summing over $d$ gives $\Theta(d^2/t)$ (multiplied by the eigenvalue dependent factors). We can include this description in the proof roadmap and intuition section near the beginning of the paper.
>
> Additionally, we can provide a brief proof sketch showing that non-adaptive compression incurs a quadratic penalty of $O((d/m)^2)$ rather than the linear $O(d/m)$ achieved with adaptation. This would be a new addition relative to the submitted version, but we believe it is important as it formally justifies the use of adaptive measurements in practice.

---

> > ### Author Rebuttal · Reviewer_fH1c · 2026-04-02
> >
> > I sincerely thank the authors for their detailed rebuttal, which has effectively addressed my concerns.

---

> > > ### Author Response · Authors · 2026-04-06
> > >
> > > Thank you for your detailed review comments. We appreciate you taking the time to read our work.

---

### Official Review · Reviewer_LrjN · 2026-03-12

**Soundness:** 3
**Presentation:** 3
**Significance:** 2
**Originality:** 3
**Overall Recommendation:** 5
**Confidence:** 4

**Summary:**

This paper gives an analysis of an algorithm for principal eigenvector estimation using adaptive sensing. At each iteration, the algorithm observes two measurements: one along the current iterate and one orthogonal to the current iterate. Using a compressed version of Oja's algorithm along these two directions, the algorithm achieves matching upper and lower bounds on estimation in the Gaussian setting. Limited synthetic experiments verify 1) better rates for adaptive versus non-adaptive measurements, 2) correct scaling with respect to dimension as predicted by the theory, and 3) the ability to track a moving eigenvector.

**Compliance With Llm Reviewing Policy:**

Affirmed.

**Final Justification:**

The rebuttal addressed my main concerns, where the authors outline a concrete application of streaming PCA that is relevant to the community. They also have helped to distinguish the theoretical contributions, and I believe this is a valuable addition to the literature.

**Key Questions For Authors:**

1. What impact do you envision compressed streaming PCA having on the broader machine learning community? What key applications might it have that drive the development of this method and make it important to know about and study in an ICML context?
2. Can you give an overview of how the upper bound proof is not just a standard analysis of using streaming stochastic PCA? What about the lower bound being a standard Assouad argument?
3. If we impose further structure on the top eigenvector (e.g., it is sparse), can we achieve rates that depend on this rather than the full dimension?

My first two questions are the most important to address, because these are my main concerns about how significant this result may be to the broader ICML context. Along with the answers, I would suggest editing the text to reflect these answers to make it more broadly accessible.

**Limitations:**

yes

**Strengths And Weaknesses:**

Strengths:
- The paper appears to be technically sound, appealing to standard techniques for bounding risk for stochastic gradient algorithms along with standard statistical techniques to prove lower bounds on statistical estimators.
- The paper is clearly written and easy to follow.
- The paper is the first to give matching upper and lower bounds for compressed eigenvector estimation.
- The work appears to be clearly original, with results that do not appear in the literature.

Weaknesses:
- In terms of significance, the algorithm is not novel. The analysis is restricted to the Gaussian setting alone. Furthermore, while PCA is a core problem in data analysis and machine learning, it is unclear how broad an impact the discussed compressed variant of PCA will have, more broadly speaking.
- The lower bound proof technique appears relatively standard. I am unsure of how novel the proof technique for the upper bound is compared to existing analyses of compressed streaming PCA.
- Empirical results are limited, which is not necessarily bad since the results are primarily the matching theoretical upper and lower bounds.

---

> ### Author Rebuttal · Authors · 2026-03-31
>
> Thanks for the thorough assessment. We agree the algorithm is not our contribution --- it comes from combining Oja's algorithm with the measurement scheme from Adaptive GROUSE. Our main contributions are presenting a coherent narrative and analysis of compressed eigenvector estimation in the noisy setting as well as matching lower bounds.
>
> **Weaknesses**
>
> We agree that the algorithm is not novel. Though our argument is restricted to the Gaussian setting, extension to general sub-Gaussian distributions is straightforward but not particularly illuminating. Constants would change depending on higher order moments as opposed to Isserlis, but the core rates for upper and lower bounds would be identical. Because PCA is such a core technique in statistical estimation, we believe extending the theory to compressive observations (first to rank-1, later to rank-k) is useful to the broader community.
>
> Further, there are numerous applications where compressive observations are necessitated by reality, particularly in modern radio frequency systems. There are prevalent problems like direction of arrival estimation or massive MIMO beamforming where the core problem boils down to estimating the principal eigenvector/subspace of a covariance matrix where our algorithm will see immediate application.
>
> The lower bound proof technique is indeed standard once the stochastic recurrence is established. We can add an appendix with more comprehensive comparisons between our upper/lower bounds and empirical trajectories.
>
> We can also provide a brief proof sketch showing that non-adaptive compression incurs a quadratic penalty of $O((d/m)^2)$ rather than the linear $O(d/m)$ achieved with adaptation. This would be a new addition relative to the submitted version, but we believe it is important as it formally justifies the use of adaptive measurements in practice.
>
> **Q1 (Impact & applications)**
>
> The most immediate applications of compressed streaming PCA are in edge devices and sensors. Modern radio frequency systems contend with a "data deluge" as the sensor counts and bandwidths increase, and leading eigenvector/subspace estimation are core problems in this setting. Recommender systems may also see benefit, since observations are inherently sparse. Our results quantify the tradeoff between compression and samples, encouraging the use of more, cheaper samples over a few costly ones.
>
> Our bounds give concrete design guidance: the $\Theta(d^2/t)$ rate decomposes as $\Theta((d/m) \cdot d/t)$, where $d/t$ is the fully-observed minimax rate and $d/m$ is the information-theoretic price of compression. Concretely, $d = 64$ with 4 RF chains converges $4\times$ faster than $d = 128$ with 2 RF chains at comparable measurement energy. We treat $m = O(1)$, $k = 1$, Gaussian as a starting point that gives sharp results with clean proofs, and view it as a foundation for the general setting (arbitrary $m$, rank-$k$, sub-Gaussian) where we expect the same $d/m$ scaling to hold.
>
> **Q2 (Proof novelty)**
>
> Once the stochastic recurrence is derived, the upper bound proof uses standard stochastic recurrence arguments. We cite Balsubramani et al. which provides a similar analysis for full-dimensional PCA, but with looser warmup bounds. We disagree that our upper bound is a direct rewrite of any stochastic PCA proof, but do agree that many of the tools are standard.
>
> The key distinction is that the compressed adaptive setting introduces a multiplicative coupling between the iterate and the measurement operator. In fully-observed Oja (e.g., Allen-Zhu & Li 2017), conditioning on $\mathbf{u}_t$ makes the noise independent of $\mathbf{u}_t$, so standard martingale arguments apply. In our setting, $\mathbf{u}_t\mathbf{u}_t^T + \mathbf{b}_t\mathbf{b}_t^T$ *is* the measurement operator --- noise and observation are multiplicatively coupled through the iterate. We instead track $c_t^2$ through a stochastic recurrence with self-term and cross-term bounds.
>
> The lower bound is a standard Assouad argument once the KL divergence is established, but the reduction to orthonormal measurement matrices and the energy budget are novel in the lower bound development. Additionally, we note that Assouad has not been applied to develop lower bounds for even full-dimensional PCA. Our choice of argument over the traditional Fano or Le Cam approaches is needed for the extension to the compressed setting.
>
> **Q3 (Sparse eigenvector)**
>
> Certainly, sparsity would achieve better rates, but we believe this extension merits its own work and would distract from the core contributions here. For $s$-sparse $\bar{\mathbf{u}}$, our energy budget restricts to $s-1$ coordinates, each getting $O(t/s)$ energy, giving $\Omega(s^2/t)$. The fully-observed sparse PCA rate is $\Theta(s\log(d/s)/t)$ (Cai, Ma & Wu, 2013), so compression replaces $\log(d/s)$ with an additional factor of $s$. A matching upper bound would require support recovery.

---

> > ### Author Rebuttal · Reviewer_LrjN · 2026-04-02
> >
> > I appreciate the authors taking the time to address my concerns. I think adding some compelling text for the applications, as discussed in their rebuttal, would be nice. I also appreciate the addition of theory that compares adaptive and nonadaptive rates, which would help to highlight how this is a valuable addition.

---

> > > ### Author Response · Authors · 2026-04-06
> > >
> > > Thank you for taking the time to read our work. The adaptive vs. non-adaptive rates was a new piece of insight we found after the original submission, so we will be happy to include it in the final revision.

---

### Official Review · Reviewer_7Ve8 · 2026-03-12

**Soundness:** 3
**Presentation:** 3
**Significance:** 2
**Originality:** 3
**Overall Recommendation:** 4
**Confidence:** 3

**Summary:**

This work studied the Principal Eigenvalue Estimation problem with only two adaptive measurements per iteration. They proved convergence rate $\mathcal O(\lambda_1 \lambda_2 d^2 / \Delta^2 t)$ with a matching information-theoretic lower bound, say $d^2$ factor is unavoidable. The Assouad-Lemma based derivation can be applied to lower bound for other PCA cases. They also provided a fixed point analysis when the eigenvector is moving.

**Compliance With Llm Reviewing Policy:**

Affirmed.

**Final Justification:**

My concerns are largely resolved and I will keep my positive score.

**Key Questions For Authors:**

- The paper claim that prior work are “with high probability in expectation” bounds, however in Balzano's paper, it's an expected improvement for the noisy case(maybe I am wrong). And their noise is assumed Gaussian - possibly in the whole space, even in the principal direction. I feel like the claim on noisy situation is not exactly the case. What if there exists real sensing noise rather than the structral signal induced by $\lambda_2 \not = 0$?
- Why we are interested in this particular case, where only 2 measurement per iteration, and limited storage resources only allowed for streaming online learning, is there any real applications for this situation. I feel like that if the front-end system complexity bound and computation complexity bound usually do not exist at the same time. Say if the data are valuable, then it should be used multiple times.
- How does the bound for rank-1 estimation in this paper compared to general rank-k subspace estimation, and only keep the top-eigenvalue, as in Randomized SVD, keeping some redundunt dimension will give better result?

**Limitations:**

- The Gaussian assumption is key for everything in the framework, but in practice, it's likely that some directions are correlated or heavy-tailed.
- For tracking a moving eigenvector, authors assume that the drift direction is not systematically aligned with or against the algorithm’s exploration, which is usually not the case as an isotropic random walk.

**Strengths And Weaknesses:**

Strength:
- The paper study a case where observations do not perfectly lie in the signal subspace (i.e., $\lambda_2 != 0$). The stochastic recurrence analysis treat the complex signal-noise interactions and achieve the minimax optimal convergence rate.
- The application of Assouad’s lemma using a measurement energy budget constraint to bound the KL divergence is elegant. By showing that total measurement energy must be distributed across $d-1$ dimensions, restricting precision to $\mathcal{O}(t/d)$ per coordinate, the authors provide a very clean way to prove the information-theoretic floor.
- Extensive experiments validate the theoretical bounds, the theoretical upper bound is clear.

Weakness:
- Limited use case for extremely restritive situation, say only two measurements per iteration and streaming update.
- While mathematically optimal for two measurements, the $d^2$ scaling makes this specific setup practically unusable for massive dimensions, actually in experiments even for d = 256, the iterations go up to 13M.

---

> ### Author Rebuttal · Authors · 2026-03-31
>
> Thank you for the review and for the kind words on the Assouad energy budget argument.
>
> **Weaknesses**
>
> We agree that the rank 1 case with 2 measurements is extremely restrictive. However, the framework and proof we have developed is a clear stepping stone towards arbitrary rank $k$ and number of measurements $m$. Further, we believe that in this simple setting, the analysis itself is illuminating as to the core dynamics of the subspace estimation problem. The lower bounds also explain the exact cost of compression as the compression factor of $d/m$. We may also provide a brief proof sketch as to why non-adaptive compression incurs a further penalty for a total detriment of $O((d/m)^2)$, which encourages the use of adaptation in practice. Performing all of this in the rank-1 setting with Gaussian data streamlines the analysis while attacking the core challenge of streaming estimation from adaptively compressed data.
>
> Prior critiques of our work were that the $d^2$ penalty seems very punishing --- we agree. But a key contribution is to show that this penalty is unavoidable if compressed data is used. Real world systems should be engineered with this compression-sample complexity tradeoff in mind.
>
> **Q1 (Prior work & sensing noise)**
>
> Our Section 4 characterization was imprecise. Zhang & Balzano (2016) does give expected improvement bounds for the noisy *fully-sampled* case; the compressive/adaptive results we should have distinguished more clearly (Zhang & Balzano 2022; Ongie et al. 2017) both assume noiseless data. We will correct the framing.
>
> On measurement/additive noise vs. structured signal $\lambda_2\neq0$: isotropic $\epsilon_t \sim N(0, \sigma_\epsilon^2 I_2)$ shifts the covariance $\Sigma \to \Sigma + \sigma_\epsilon^2 I$, leaving the spectral gap $\Delta$ unchanged, so the $d^2/t$ rate is preserved. For anisotropic measurement noise, as long as the principal direction remains unchanged, our analysis generalizes to this setting as well.
>
> **Q2 (Applications)**
>
> The 2-measurement restriction is a *physical/hardware* constraint --- massive MIMO systems have $d = 64$--$256$ antennas but only 2--4 RF chains. To put the iteration count in context: at a sampling rate of just 20 MHz, 13M iterations for $d = 256$ is just 0.65 seconds of wall-clock time. Streaming is necessitated by the "data deluge" seen in modern RF systems. Ultimately, we do not intend to just settle on the 2 measurement case and aim to generalize to arbitrary rank-$k$ with $m$ observations, but there are notable settings where rank-1 estimation is important including single Direction of Arrival (DoA) estimation, Massive MIMO beamforming (estimating the principal eigenvector of the channel covariance), or community detection (with 2 communities, need to find the leading eigenvector of the centered adjacency matrix).
>
> **Q3 (Rank-$k$)**
>
> Our lower bound applies to *any* estimator using 2 measurements, including one that estimates rank-$k$ and extracts the top eigenvector. Randomized SVD requires full-dimensional samples, so it does not apply here. The bottleneck is the measurement budget $m$, not the target rank; for general $m$, the energy budget gives $\Omega(\lambda_1\lambda_2 d^2/(m \Delta^2 t))$. Choosing $m=O(1)$ in our case yields the presented result.
>
> **Q4 (Limitations)**
>
> On Gaussianity --- the $d^2/t$ rate should extend to sub-Gaussian data, since Isserlis' theorem (upper bound) can be replaced by sub-Gaussian moment bounds, and rotational invariance (lower bound) by Le Cam's method or similar. On tracking drift: without a dynamics model, the fixed-point analysis (Eq. 5) gives the best achievable tradeoff between tracking speed and estimation noise. Exploiting structured drift would require a state-space model, which is a qualitatively different problem, but could conceivably reduce the fixed point error.

---

> > ### Author Rebuttal · Reviewer_7Ve8 · 2026-04-03
> >
> > Thank you for the comprehensive response. My concerns are largely resolved. The explanation on "data deluge" seen in modern RF systems do provide application cases for 2 channel but streaming algorithm.

---

> > > ### Author Response · Authors · 2026-04-06
> > >
> > > Thank you for your review. We agree that specifying existing applications for rank-1 and rank-k streaming algorithms is useful to frame the research contribution for the broader ICML community. This will be done in the revision.

---

### Decision · Program_Chairs · 2026-04-30

**Decision:**

Accept (regular)

**Comment:**

The paper studies PCA under compressed sensing in the following sense:
at each timestep, instead of observing a fresh sample $x_t$ from the underlying distribution,
the algorithm observes only $A_tx_t$ for a $2 \times d$ matrix $A_t$. Here, the matrix $A_t$ is chosen by the algorithm designer. That is, at each step, the algorithm observes only two real-valued linear measurements. Given such restricted measurements, the goal is to identify the top eigenvector of the covariance of $x_t$.
The paper proves that the optimal rate of estimating the top eignvector in the sin-squared metric is equal to $\Theta(d^2/t)$, at least for the constant eigengap and constant top two eigenvalues. The upper bound holds for a particular choice of adaptive sensing matrices $A_t$, whereas the lower bound holds for any choice.

The reviewers agree that the paper essentially resolves this problem by establishing matching upper and lower bounds, which constitutes an interesting technical contribution. The paper is also well written. Overall, the paper would likely be of interest to the ICML community. However, there are some reservations regarding the use case and the somewhat limited scope of the setting, and it would be beneficial to provide stronger motivation.

Further discussion on the role of adaptivity and distributional assumptions would also be valuable. In particular, could one attain the same rate using non-adaptive sensing matrices $A_t=A$, or are there provably stronger lower bounds in the non-adaptive setting? Additionally, how essential is the (sub)-Gaussian assumption for the upper bound, and would the same rate hold under heavy-tailed distributions?